# The genetic and environmental composition of socioeconomic status in Norway

Joakim Coleman Ebeltoft [1] ✉, Espen Moen Eilertsen[1], Rosa Cheesman[1], Ziada Ayorech [1], Arno Van Hootegem [2,3], Torkild Hovde Lyngstad [2] & Eivind Ystrom [1,4,5]

Estimating the contributions of genetic and environmental factors is key to understanding differences in socioeconomic status (SES). However, the heritability of SES varies by measure, method, and context. Here, we estimate genetic and environmental sources of variance and commonality in the 'big four' SES indicators. We use high-quality administrative data on educational attainment, occupational prestige, income, and wealth, and employ four family-based and unrelated genotype-based heritability methods, all drawn from the same population-wide cohort of >170,000 Norwegians aged 35-45. By drawing subsamples from a consistent sample and using registry-based data, we reduce differences in estimates due to population characteristics and measurement error. Our results show that genetic variation consistently explains more for educational attainment and occupational prestige. Family-shared environmental contributions explained more for educational attainment and wealth. Our results highlight considerable common influences on the four SES indicators among genetic and shared environmental factors, but not among non-shared environmental factors. Overall, we show how the relative importance of genetic and environmental factors to SES differences in Norway varies by method and type of socioeconomic attainment. This study is a reliable source for comparing heritability methods, and for comparing SES indicators and their genetic and environmental commonality in a social-democratic welfare state.

It is well established that socioeconomic status (SES) is connected to an array of important life outcomes, such as health and mortality[1], and subjective well-being[2]. Given the significance of SES, extensive literature has tried to understand the shared environmental and genetic sources of variation in SES. Despite being a contentious topic with a complex history (elaborated upon in Ethical and societal considerations in Discussion), research on the relative magnitude of genetic and environmental

contributions to SES has provided coarse-grained but important insights into the complex interplay of mechanisms creating variation in socioeconomic outcomes. Although the field is advancing, significant challenges remain.

While there is broad agreement on the significance of SES, there is less consensus on how to operationalize and measure the construct across multiple disciplines[3–6]. For example, in a recent literature review, SES was represented using 149 unique indicators[4]. Each of the

[1]PROMENTA Center, Department of Psychology, University of Oslo, Oslo, Norway. [2]Department of Sociology & Human Geography, University of Oslo, Oslo, Norway. [3]Centre for Fertility and Health, Norwegian Institute for Public Health, Oslo, Norway. [4]PsychGen Centre for Genetic Epidemiology and Mental Health, Norwegian Institute of Public Health, Oslo, Norway. [5]Centre for Research on Equality in Education, Faculty of Education, University of Oslo, Oslo, Norway. ✉e-mail: joakimeb@uio.no

indicators captures a specific aspect of socioeconomic variation, while also reflecting a common status dimension[5,7–10]. Researchers typically study this commonality by calculating an SES composite index based on several indicators[11]. Most operationalizations of SES make use of at least one or more of the 'big four' SES indicators—educational attainment, occupational prestige, income, and wealth[4,12]. However, a systematic test and comparison of their shared or specific environmental and genetic sources of variation on the 'big four' remains largely lacking.

A wide range of methods are employed to estimate heritability, the proportion of phenotypic variance attributed to the cumulative effect of genetic differences. The approaches are historically divided into, first, family-based methods that infer genetic associations by applying expected relatedness between twins and other family members, and, second, genotype methods that estimate genetic associations between unrelated individuals using empirical variations in single-nucleotide polymorphisms (SNP) in the genome. There is often a gap between family-based and unrelated genotype-based heritability estimates, referred to as 'missing heritability.' This gap has also been demonstrated in genetically informed studies of SES. As shown in Fig. 1, family-based studies of the 'big four' find heritability estimates ranging from 0.29 to 0.61[13–18], while unrelated genotype-based studies find estimates ranging from 0.07 to 0.21[19–23]. However, these estimates have been obtained from different populations. To assess the severity of the 'missing heritability' problem for SES, we need to compare samples that share similar population characteristics.

Heritability is a population statistic that is conditional on the features of the population and inherently reflects the characteristics of the populations we study. For example, societal features, such as a free and open higher education system, may yield higher heritability of educational attainment[24]. As the different colors in Fig. 1 indicate, differences due to country[13,16], cohort[25], age[16], and selection bias[26] affect heritability estimates, adding to the difficulty of comparing SES indicator estimates across populations. Large genome-wide association studies (GWAS) results for SES have enabled heritability to be estimated in new ways[21,22,27]. However, these studies usually combine a large number of samples from heterogeneous contexts. This potentially washes out population-specific signals and makes comparisons difficult across SES indicators (e.g., estimates based on income and education GWAS) and methods (e.g., estimates based on the education GWAS versus a twin study of education). To assess differences in heritability across SES indicators and methods, the use of a consistent sample is needed.

Furthermore, in the SES heritability literature, there has been an over-reliance on self-reported measures. Many factors can distort self-assessed measures ranging from misunderstanding of questions to memory errors and social desirability biases. In the UK Biobank, which is the basis of much genomic research on SES, measurement repeatability was 0.78 for educational attainment and 0.46 for income[28]. Objective measures are needed for reliable comparison between SES indicators and heritability methods.

Wealth is significantly understudied particularly compared to educational attainment and income. This gap in the literature is likely due to a lack of reliable individual-level data on wealth. This is important because wealth is the most unequally distributed SES indicator world-wide[29], and even in 'egalitarian' and strong universal welfare states such as Norway[30]. The only genotype-based study had a relatively small sample size of ~8000[31]. Multivariate family-based genetic studies, including wealth, have thus far not been published. Genetically informed studies using reliable measures of wealth are

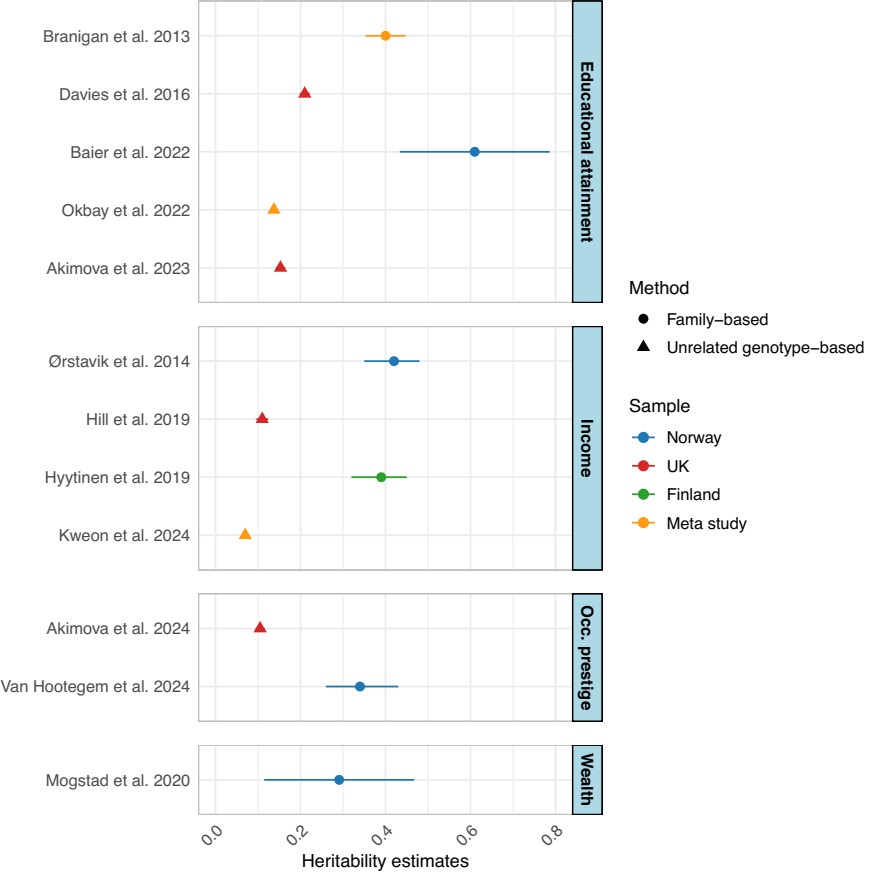

**Fig. 1 | Selection of heritability estimates in current literature.** Heritability estimates on educational attainment, income, occupational prestige, and wealth from European and meta-study samples. The points represent the heritability estimates, and the lines have 95% confidence intervals. Sample size varies from study to study.

**Table 1 | Range of genetic correlations for the 'big four' in a selection of the current literature**

| SES indicator | Educational attainment | Occupational prestige | Income | Wealth |
|---|---|---|---|---|
| Educational attainment | 1 | 0.73[31]–0.96[23] | 0.81[23]–0.92[22] | 0.82[31] |
| Occupational prestige | 0.81[17] | 1 | 0.91[23] | 0.60[31] |
| Income | 0.52[17] | 0.75[17] | 1 | N/A |
| Wealth | N/A | N/A | N/A | 1 |

Family-based genetic correlations are in the lower triangle, and unrelated genotype-based correlations are in the upper triangle.

**Table 2 | The four applied methods and their key features**

| Design | Key features |
|---|---|
| Family pedigree (FP) | The basic model assumes genetic correlation between siblings ($r_g = 0.5$) and first cousins ($r_g = 0.125$) and shared environment correlations between siblings ($r_c = 1$) and first cousins ($r_c = 0$). Similarity that is not accounted for by shared environment correlations implies genetic effects. |
| Identity-by-descent (IBD) | Uses empirical genetic correlations among full siblings. The extent to which phenotypic correlations match genotypic correlations implies the degree of heritability. |
| Genome-based restricted maximum likelihood (GREML) | Uses empirical genetic correlations among unrelated individuals. Phenotypic correlations that match the pattern of genotypic correlation imply higher heritability. |
| Linkage disequilibrium (LD) score regression | Uses GWAS summary statistics and estimates SNPs associations with outcomes by correlating linkage disequilibrium scores with Chi square statistics. |

For further description, see the Method section under Heritability Estimation.

needed to shed light on how genetic variation contributes to the intergenerational transmission of wealth.

A crucial question in the familial origins of SES is the structure of the influences on the different outcomes. Studies suggest strong overlap, with genetic correlation ranging from 0.52 to 0.96 (Table 1)[17,22,23,31]. However, the issues discussed above, such as consistent samples, methods, and self-report bias, prevent us from accurately establishing the genetic structure of SES. The genetic and environmental covariate structure of SES indicators is important as it can shed light on what the commonly used composite SES indicators consist of. In sum, there is a need for multivariate studies on the 'big four' SES indicators in a consistent sample, using a variety of methods and objective measures.

In response to the challenges described above, we conducted our analyses on a homogeneous sample of Norwegian adults aged 35 to 45. First, this approach allows us to compare the relative importance of genetic and environmental factors for the 'big four' in Norway, a social-democratic universal welfare state. We apply four different heritability estimation methods, two family-based and two unrelated genotype-based (Table 2), drawn from a consistent sample (see Supplementary Data S1), to enable comparison of estimates across methods. Second, we evaluate the commonality of genetic and environmental factors on the 'big four' by performing multivariate covariance analysis and applying a dimensionality-reduction technique. For all analyses, we draw subsamples from the same population-wide cohort of parents largely born in post-welfare Norway and use yearly objective registry data from ages 35 to 45. As we use rich registry-based data with low missingness, we have similar sample sizes across SES indicators (see Methods section for details). The research questions in the current study are:

1. What are the genetic, shared environmental, and non-shared environmental contributions to SES indicators in a Norwegian context?
2. How much do 'big four' heritability estimates vary between family-based and unrelated genotype-based methods?
3. How much of the genetic, shared environmental, and non-shared environmental contributions are shared across educational attainment, occupational prestige, income, and wealth?

We conducted our analysis on a Norwegian sample, and there are several elements to be aware of when studying genetic and environmental variation in Norway. First, Norway has systems with social safety nets that reduce the impact of negative life events on SES, and create opportunities for second chances in socioeconomic attainment[32]. Second, Norway and other social-democratic welfare states possess features that reduce the importance of certain environmental factors[33]. For example, universal high-quality preschool systems relieve the burden of financial resources as well as time that can be spent in the labor market. Third, intergenerational mobility is relatively high in Norway, partly due to its social-democratic welfare state that provides universal access to social protection and higher education[32,34]. Higher intergenerational mobility has been found to be positively correlated with heritability[35]. One interpretation of this correlation is that when differences created by environmental barriers or opportunities (such as the cost of higher education) are minimized, genetic differences account for relatively more variation in outcomes[36]. Fourth, despite the strong level of educational and income equality in Norway, wealth disparities are relatively high and have increased over time[37,38]. Lastly, the availability of high-quality, yearly Norwegian register data (Table 3) enhances the accuracy of findings as it eliminates biases associated with self-reported questionnaire measures[28].

## Results

### Comparing the estimates across heritability methods and SES indicators

We applied four methods (Table 2): Family Pedigree (FP)[39], Identity-By-Descent (IBD)[40], Genome-based Restricted Maximum Likelihood (GREML)[41], and Linkage Disequilibrium Score Regression (LD score regression)[27]. FP ($n = 77,285$) and IBD ($n = 22,982$) rely on family members, whereas GREML ($n = 36,277$) and LD score regression ($n = 128,310$) rely on unrelated individuals (for further detail, see Methods). Heritability estimates were statistically significant for all measures and all methods (Fig. 2 and Supplementary Table 2). Across methods, genetic contributions were highest for education (e.g., LD score regression method estimates 20% (S.E. = 0.8%) versus 6.8% for income (S.E. = 0.5%)). The two family-based methods resulted in higher heritability estimates than the two unrelated genotype-based methods, with the average FP ACE model estimate across indices of 32% and the average GREML estimate across indicators of 15%. This study provides novel estimates of the heritability of wealth, finding similar family-based estimates to income (FP ACE model estimates were 25%

**Table 3 | The 'big four' SES indicators and how they were measured in the current study**

| SES indicator | Current measure |
|---|---|
| Educational attainment | Data from Norwegian National Educational Database was formatted in the International Standard Classification of Education (ISCED) 2011 and converted to the number of years required in Norway to obtain each category. We used the highest educational attainment registered between the ages of 35 and 45. |
| Occupational prestige | Occupational codes from Statistics Norway were converted to Treiman's Standard International Occupational Prestige Scale (SIOPS)[66]. SIOPS has been shown to be relatively stable across contexts and time points and has been widely used in social mobility research[6,67,68]. We averaged occupational prestige across an 11-year period from age 35–45. |
| Income | Statistics Norway's cross-referenced registry-based measure of an individual's total income after taxes, which consists of wages, capital income, and taxable and non-taxable transfers after taxes during a calendar year. We used log-transformed and averaged income across an 11-year period from age 35 to 45. |
| Wealth | Statistics Norway's cross-referenced registry-based measure of an individual's gross wealth, which is a sum of real capital and estimated financial capital, i.e., all the financial resources a person legally has tied to their name. We used log-transformed and averaged wealth across an 11-year period from age 35 to 45. |

For further description, see the Method section under Measures.

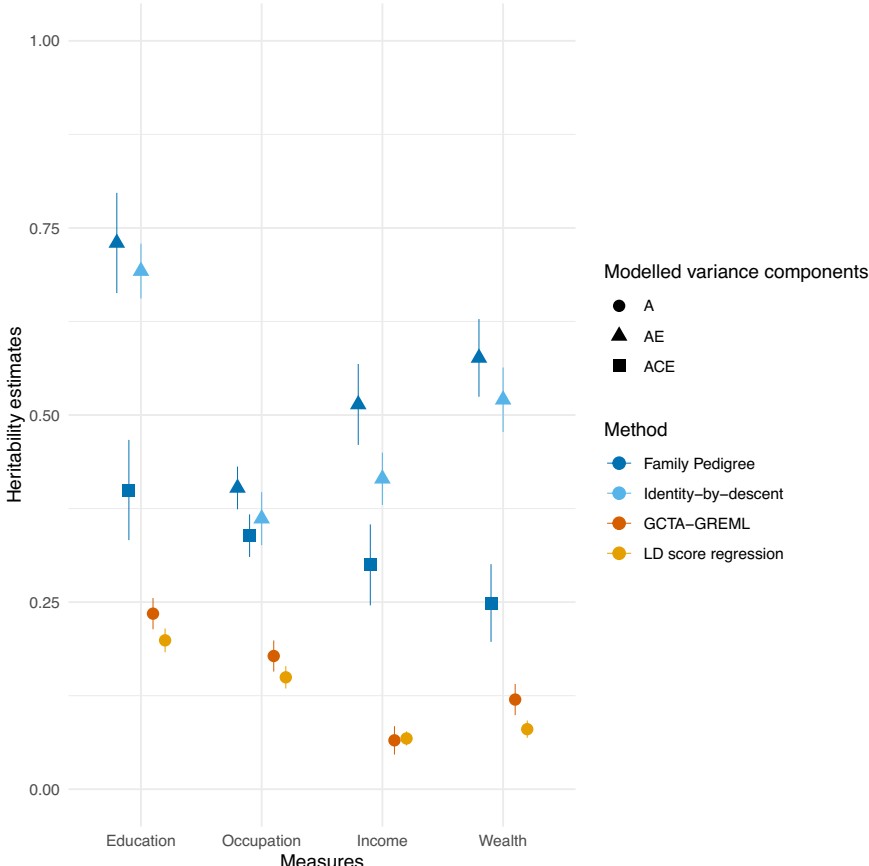

**Fig. 2 | Heritability estimates.** Heritability estimates for education, occupation, income, and wealth across the heritability methods Family Pedigree, Identical-by-descent, Genome-wide complex trait analysis genome-based restricted maximum likelihood (GCTA-GREML), and Linkage Disequilibrium (LD) score regression. The points are the estimates, while the lines are 95% confidence intervals. See Fig. 6 for sample sizes.

for wealth (S.E. = 2.7%) and 30% for income (S.E. = 2.8%)). However, the unrelated genotype-based heritability estimates for wealth were higher than for income (GREML estimates were 12% for wealth (S.E. = 1.1%) and 6.5% for income (S.E. = 1%)). We also found that FP-based heritability estimates were sensitive to the assumptions underlying the shared environment correlations (wealth was attenuated from 56% (S.E. = 1.4%) to 25% (S.E. = 2.7%) when changing the cousin shared environment correlation from 0 to 0.59).

## Extracting shared environment contributions
Family members share both genetics and environments, making it challenging to discern the sources of similarity in genetic designs that include relatives. The FP method had a large enough sample size ($n = 77,285$) to estimate a shared environment effect, while the IBD design did not ($n = 22,982$). When shared environmental effects (C) were modeled in the FP design, additive genetic effects (A) were attenuated. However, shared environmental estimates depend on the specific assumptions of the model (Fig. 3). The first assumption is no shared environment effects. The second is the standard assumption of a shared environment between siblings correlating fully ($r_c = 1$) and being uncorrelated for first cousins ($r_c = 0$) (triangles in Fig. 3). For the third set of assumptions, we set correlations between the shared environment of full siblings and twins ($n = 27,862$) to 1, while the $r_c$ between first cousins ($n = 49,423$) was a free parameter (squares in

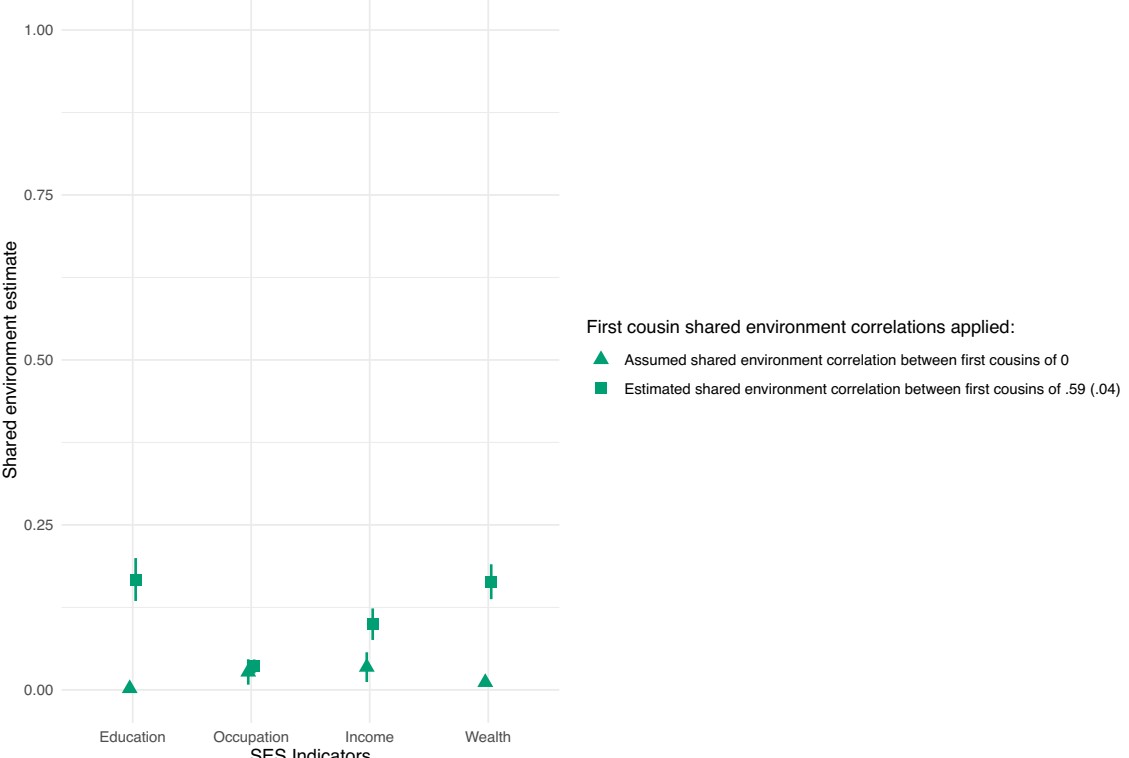

**Fig. 3 | Estimated shared environment contributions.** The estimated shared environment contributions (and their 95% confidence intervals) to education, occupation, income, and wealth were modeled with a Family Pedigree ACE design with a set sibling shared environment correlation of 1 and an estimated cousin shared environment correlation of 0.59. As with heritability, the estimates of explained variance are standardized to a total variance of 1. See Fig. 6 for sample sizes.

Fig. 3). Here, the shared environment correlation between first cousins was estimated to be 0.59 (SE = 0.04). Figure 2 shows how a model capturing the empirical shared environment correlation between cousins leads to attenuated FP heritability estimates. Figure 4 shows how sensitive the estimates are to the underlying assumptions. In Fig. 4 the estimated additive genetic, shared environment, and non-shared environment influences on educational attainment vary substantially across a full range of assumed first cousin shared environment correlations. Similar figures to Fig. 4 for occupational prestige, income, and wealth can be found in the supplementary information Figs. S5–S7.

**Examining the genetic and environmental correlations between the 'big four' indicators**

We examined the underlying structures of genetic and environmental influences on the 'big four' SES indicators. As shown in Table 4, we estimated phenotypic, genetic, and environmental correlations among indicators. Our results showed phenotypic correlations ranging between 0.24 (S.E. = 0.002) for education and income and 0.58 (S.E. = 0.002) for education and occupational status. The genetic correlations between education, occupational status, income, and wealth ranged from 0.35 (S.E. = 0.05) to 0.93 (S.E. = 0.01) for family-based designs and from 0.71 (S.E. = 0.06) to 0.96 (S.E. = 0.02) for unrelated genotype-based designs. Shared environment correlations were all 1. Non-shared environment correlations ranged from −0.12 (S.E. = 0.03) to 0.30 (S.E. = 0.02).

**The dimensionality of genetic and environmental contributions**

Another way to analyze the commonalities between SES indicators is to apply Principal Component Analysis (PCA), which reduces the dimensionality while retaining as much information as possible[42]. Parallel analysis suggested retention of one principal component (PC)

for phenotypic, genetic and shared environment variance components, and two components for the non-shared variance component (see Supplementary Figs. S8–S16)[43,44] Fig. 5 shows the explained variance for phenotypic variance component and the retained components from the IBD additive genetic and non-shared environmental components. The high variance explained by the first additive genetic PC is contrasted with the low explained variance of two non-shared environments (Fig. 5). The first two IBD E PCs explained 28.7 to 31.1%, which is 3.7 to 6.1% more than the 25% we would expect if the indicators were uncorrelated. The two retained PCs showed a distinct pattern with respect to their loadings (Supplementary Table S16). The first PC loadings were strongly positive for wealth and occupational prestige, but null to weak for education and income. In contrast, the second PC had a strong positive loading on education and a strong negative loading on income, with weak loadings on wealth and occupation. The explained variance of the unrelated genotype-based PCs exceeded 85% (see Supplementary Table S15). For the FP method, all the variance was explained by the shared environment first PC (see Supplementary Table S16). One PC was retained for the FP non-shared environmental and explained 38% of the variation. The genetic, shared environmental, and FP non-shared environmental first PCs all had similar loadings to all four indicators. Loadings of all retained PCs and IBD E retained PC circumplex can be found in the supplementary Fig. S17.

## Discussion

We estimated genetic and environmental contributions to SES in Norway. By maintaining consistent sampling and utilizing registry data, we robustly compare estimates across methods and SES indicators. We found that the 'missing heritability' gap across SES indicators was attenuated when shared environment effects were included, with a shared environment correlation between first cousins estimated from

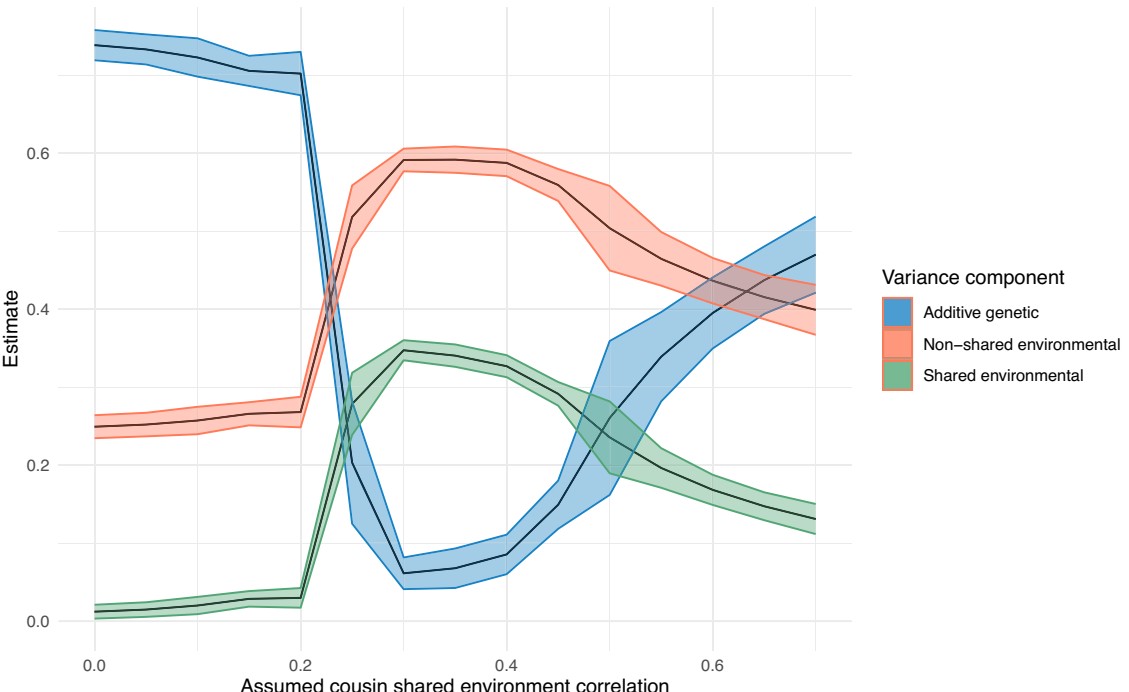

**Fig. 4 | A, C, and E contribution to educational attainment under different shared environment correlation assumptions.** Additive genetic, shared environmental, and non-shared environmental effects on educational attainment across assumptions of first cousin shared environment correlation. *X* axis represents the degree of assumed first cousin shared environment correlations. Sibling shared environment correlation was assumed to be 1. Intervals between assumptions are 0.05, and ribbons represent 95% confidence intervals. See Fig. 6 for sample sizes.

real data. There was consistently a higher genetic contribution to education compared to occupation, income, and wealth across methods. The phenotypic, genetic, and shared environmental variance of educational attainment, occupational prestige, income, and wealth appeared to be unidimensional. In contrast, the non-shared environmental variance exhibited a low degree of commonality. Specifically, individual-specific environments associated with longer education were linked to lower income.

Across the four family-based and unrelated genotype-based methods, education had the highest and occupation the second highest heritability. The similarity across methods highlights the validity of the finding in the Norwegian population. The pattern resembles the unrelated genotype-based findings in the UK, whereby the highest genetic contribution is for education, the lowest is for income, with occupation between the two[19,20,23]. Adding to the literature on the heritability of the 'big four,' this is the first time that a large genomic study on wealth has been conducted. The unrelated genotype-based heritability estimates for wealth are close to, but slightly higher than, those for income. Estimated genetic correlations suggest both overlapping and distinct processes underpinning the attainment of income and wealth, corresponding to registry data research showing that income makes up a large proportion of wealth variation in Norway (Black et al.[45]). Further investigation might shed light onto whether the genetic signal is capturing within or between family differences, and if other measures, such as applying net wealth instead of gross wealth, would yield different results[46].

Several factors could account for the 'missing heritability.' These include rare genetic effects not captured by SNP-based methods, and shared environments inflating the family-based estimates. By relying on unrelated individuals, the genotype methods circumvent the need to make assumptions regarding the sharing of environments, which are necessary in the family-based design. However, these unrelated genotype-based methods are population estimates and do not distinguish between 'direct' offspring genetic effects and 'indirect' parental

genetic effects that affect offspring phenotype through the environment[47]. Recent methodological development has focused on disentangling 'direct' genetic effects from 'indirect' genetic effects and population stratification. Within-sibship GWAS performed in the UK Biobank show that 'direct' estimates were attenuated from the 'population' estimate and decreased 76% for educational attainment[48], 74-76% for income[22], and 50% for occupational prestige was[23]. Trio-GWAS meta-analysis showed an attenuation of 49% for educational attainment, 67% for individual income, and 60% for household income[49]. We, therefore, expect that our unrelated genotype-based estimates are inflated due to indirect genetic effects and population stratification.

Although individuals with similar genetics may have similar outcomes, there is no straightforward causal relationship between genetics and SES. There are three main interpretive points to consider. First, genetic factors do not determine socioeconomic outcomes[50]. Heritability does not imply immutability, and the variance explained does not equal the range of phenotypic potential afforded by one's genetics. To the extent that genetic factors contribute to SES, they do so probabilistically through gene-environment correlations and interaction mechanisms. For example, as women's opportunities for education strengthened, the gender differences in the relationship between genetics and education weakened[51]. In Norway, genetic influences on educational performance are moderated by schools[52]. Proximal and distal social factors—ranging from parents to institutions and macroeconomic events—play crucial roles in shaping individual socioeconomic outcomes through cumulative facilitation or suppression based on genetically influenced traits. A second pitfall is interpreting heritability as a fixed value. Cultural, economic, and political changes are constantly occurring, making heritability estimates a snapshot that can vary over time and context. Lastly, heritability is a population statistic and cannot be applied to individuals.

We found the highest shared environmental contributions for wealth and education, followed by income and occupation. This may

**Table 4 | Phenotypic, genetic, shared environmental, and non-shared environmental correlations across all four methods**

| | Education | Occupation | Income |
|---|---|---|---|
| **Phenotypic correlation** | | | |
| Occupation | 0.58 (0.002) | | |
| Income | 0.24 (0.002) | 0.38 (0.002) | |
| Wealth | 0.27 (0.002) | 0.29 (0.002) | 0.37 (0.002) |
| **Family pedigree, genetic correlation** | | | |
| Occupation | 0.47 (0.03) | | |
| Income | 0.35 (0.05) | 0.59 (0.03) | |
| Wealth | 0.90 (0.02) | 0.58 (0.03) | 0.39 (0.05) |
| **Identical-by-descent, genetic correlation** | | | |
| Occupation | 0.59 (0.03) | | |
| Income | 0.59 (0.02) | 0.73 (0.03) | |
| Wealth | 0.93 (0.01) | 0.69 (0.03) | 0.66 (0.03) |
| **GCTA-GREML, genetic correlation** | | | |
| Occupation | 0.96 (0.02) | | |
| Income | 0.71 (0.06) | 0.79 (0.05) | |
| Wealth | 0.92 (0.04) | 0.79 (05) | 0.92 (0.04) |
| **LD score regression, genetic correlation** | | | |
| Occupation | 0.95 (0.04) | | |
| Income | 0.81 (0.04) | 0.90 (0.05) | |
| Wealth | 0.83 (0.04) | 0.82 (0.05) | 0.83 (0.06) |
| **Family pedigree, shared environmental correlation** | | | |
| Occupation | 1 (SE < 0.001) | | |
| Income | 1 (SE < 0.001) | 1 (SE < 0.001) | |
| Wealth | 1 (SE < 0.001) | 1 (SE < 0.001) | 1 (SE < 0.001) |
| **Family pedigree, non-shared environment correlation** | | | |
| Occupation | 0.07 (0.02) | | |
| Income | 0.05 (0.03) | 0.16 (0.02) | |
| Wealth | 0.30 (0.02) | 0.28 (0.01) | 0.12 (0.02) |
| **Identity-by-descent, non-shared environmental correlation** | | | |
| Occupation | −0.01 (0.03) | | |
| Income | −0.12 (0.03) | 0.11 (0.02) | |
| Wealth | 0.08 (0.4) | 0.23 (0.02) | −0.02 (0.03) |

Standard errors (SE) in parenthesis.

be indicative of parental effects or broader dynastic social processes[53,54]. Notably, the results were sensitive to the assumed shared environmental correlation for cousins. When we used the empirically observed correlation of 0.59 rather than the typical assumption of zero, we observed a dramatic increase in the variance explained for three of the four indicators. By refining the understanding of how much environmental factors are shared among extended family relationships, the significance of the shared environment was substantially highlighted. Interestingly, results did not differ for occupational prestige and were lower than shared environment estimates of occupational prestige from a Norwegian twin study[17]. Future work should aim to refine and cross-check approaches for modeling environmental influences shared between family members.

As with heritability, shared environmental factors need to be interpreted cautiously. First, shared environmental influences are subject to societal features. For example, labor market characteristics, educational policies, social programs, and redistribution policies all influence the role of shared environments in SES differences. Second, interactions between additive genetics and shared environments are encompassed within the additive genetic variance in these types of designs[55]. An example of an A-by-C interaction is a study that showed

the polygenic index for educational attainment interacted with the level of parental education at different stages of educational attainment[56].

The large genetic commonality aligns with prior findings on overlap in genetic factors between indicators of SES[17,22,23,31]. We also found a unidimensional shared environment structure, although it is difficult to discern if the 100% correlation between indices is a methodological artifact or if influences shared between siblings and cousins are completely the same across the 'big four.' Both family-shared variance components, additive genetic and shared environment, displayed large first PCs that had equal loadings to each of the 'big four' SES indicators. This indicates that the family-shared contributions confer similar influences on all four, whereas environments not shared differentiate chances of socioeconomic attainment between education, occupation, income, and wealth. However, with only four input variables, the variation and number of possible identified dimensions are limited. More fine-grained and horizontally differentiable indicators such as choice of educational field[57] and workplace characteristics[58] would possibly add more information[57].

A significant part of the variation in SES is explained by non-shared environmental influences[59]. Even though nonshared environmental influences are generally characterized by random chance[60], our results show that multivariate covariance structure analysis may reveal some systematicity. Although the distinct patterns of wealth and occupation loading on one PC and education and income loading on the other might be due to collider bias, there might be structural differences that have systematic effects on pathways toward status. Other social forces might be peers, such as partners, friends, and colleagues, that affect one sibling and not the other, creating differences in value hierarchies and pathways to status.

These results suggest that the common approach of studying a single composite of SES likely fails to capture a substantial proportion of individual-specific effects. This has implications for how we apply and interpret the widely used univariate composite SES indices, a practice prevalent across many fields of research[61]. Employing a composite index that only captures genetic and shared environmental commonalities might be useful; however, such use should be an informed and deliberate choice.

The current study has several limitations. First, it has limited generalizability. Educational attainment, occupational prestige, income, and wealth are upwardly biased in the current sample compared to the broader population (see Supplementary Data S1). Therefore, the estimates might be more applicable to a middle-class to upper-middle-class Norwegian population. Future research should address the bias introduced by nonrandom study participation[26]. The use of only European ancestries also significantly restricts the study's generalizability. Second, due to the inherent properties of the designs, such as the reliance upon related *or* unrelated individuals, the same subsample cannot be used for all four methods. Supplementary Data S1 shows slight differences between the subsamples, yet these are far smaller than those between the overall population and the main sample. Third, our estimates are not causal. Several confounders are likely involved, including indirect genetic effects and population stratification. Even though we controlled for PCs, they do not sufficiently capture recent stratification, an important factor when studying SES attainment within a high-mobility context[49].

Overall, the use of registry data and a homogeneous sample allows our results to serve as a point of reference for researchers interested in the genetic and environmental contributions to SES, particularly when comparing methodologies and SES indicators. To understand the relatively high heritability estimates for education and occupational prestige, we must consider the environment under investigation: Norway's robust universal welfare state with free education and a comprehensive safety net. Although observational and non-causal, our study provides insights into the extent of the

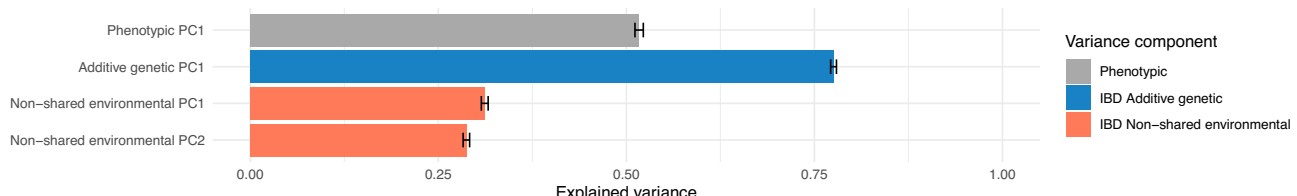

**Fig. 5 | Explained variance of principal components.** Explained variance of 1st phenotypic principal component (PC), and from the 1st additive genetic (A), and the 1st and 2nd non-shared environmental (E) principal components from the Identity-by-descent (IBD) design. 95% confidence intervals are shown as error bars. See Fig. 6 for sample size of the IBD design.

connection between genetic variation and SES in Norway. Future research could contribute to the literature on the differences in the role of genetics across various contexts. Just as we cannot fully comprehend heritability without context, we cannot fully understand the environment without considering genetics. Our results are relevant to a broader audience interested in SES and the role of shared environments, demonstrating relatively larger shared environmental effects for educational attainment and wealth. Furthermore, the findings have implications for how we fundamentally understand and model SES, showing that the common approach of operationalizing SES as a single composite index fails to capture individual-specific influences.

### Ethical and societal considerations

The nature-nurture debate has been and will remain contentious. The history of genetic research is tainted by the eugenic movement and linked to human rights abuses. It is crucial to acknowledge this history while also recognizing that genetically informed tools can help us better understand complex relationships such as those between socioeconomic inequalities and health. Appropriate interpretation of heritability and environmental estimates requires both scientific and ethical sensitivity.

In this paper, we estimate the genetic and environmental variance components of four aspects of SES in Norway. Heritability estimates reflect additive genetic effects but do not specify the mechanisms of these effects, which are likely to operate via environmental channels[62]. Interpretation of heritability estimates requires well-specified statistical models. For example, if genes and environments do not combine additively but mainly interact to shape individual differences, the variance component model is inappropriate and may result in misleading inferences regarding the importance of genes and environments.

Genetic determinism, essentialism, and reductionism are common interpretational pitfalls with ethical implications[50]. First, genetic factors do not determine but probabilistically influence SES outcomes in concert with social factors. In Norway, due to free and open education and universal study stipends and loans, genetic factors may explain more of the variance in educational attainment than in contexts where environmental factors such as family education, networks, and wealth restrict educational opportunities. Not just historically but recently, genetic determinism has been misused to support atrocities such as eugenics and racially motivated violence[63]. Emphasizing that genetic factors do not determine people's socioeconomic outcomes is therefore paramount. Instead, the importance of genetic factors depends on the environment in which they are expressed. Second, an individual's genetic 'endowment' for SES does not constitute their essence or 'underlying nature.' Some research suggests that essentialist beliefs can lead to prejudice. Third, reductionism is the misinterpretation that our genetic makeup can mechanistically explain higher-level phenomena like SES. A classic example is the 'red hair hypothetical'[64]. If children with red hair are not allowed in school, their literacy will be worse, and the genes that determine red hair will, in this specific societal context, be associated with illiteracy. Understanding the biology of red hair color would give reductive answers to the socially constructed literacy inequalities.

## Methods

### Sample

The data is from the Norwegian Mother, Father, and Child Cohort Study (MoBa), a prospective population-based pregnancy cohort study conducted by the Norwegian Institute of Public Health. Pregnant women from across Norway were recruited between 1999 and 2008, with 41% of all pregnant women participating. The cohort includes approximately 114,500 children, 95,000 mothers, and 75,000 fathers. MoBa has been linked with Norwegian registry data provided by Statistics Norway. Version 12 of the quality-assured MoBa data files, linked with registry data collected between 1960 and 2018, was used. The current study is based on registry data for MoBa parents ($n$ = 170,202). We gathered measures of education, income, wealth, and occupation from Statistics Norway registry data linked to the MoBa parents. The data are of high quality and not subject to attrition.

### Ethics

The establishment of MoBa and initial data collection was based on a licence from the Norwegian Data Protection Agency and approval from The Regional Committees for Medical and Health Research Ethics. The MoBa cohort is now based on regulations related to the Norwegian Health Registry Act. The current study was approved by The Regional Committees for Medical and Health Research Ethics (project # 2017/2205). The project has undergone review by independent ethics advisors appointed by the European Research Council (Grant agreement No. 101045526).

### Genotype quality control

Blood samples were obtained from both parents during pregnancy and from mothers and children (umbilical cord) at birth. Quality-controlled genotyping array data for the full 207,569 unique MoBa participants was recently generated[65]. Phasing and imputation were performed with IMPUTE4.1.2_r300.3, using the publicly available Haplotype Reference Consortium release 1.1 panel as a reference. To identify a sub-population of European-associated ancestry, PCA was performed with 1000 Genomes phase 1 after LD-pruning. The number of adults with quality-controlled genotype data used in this study is 128,310.

### Measures

Four measures of SES were used: educational attainment, occupational prestige, income, and wealth. The flowchart in Fig. 6 details the data preparation process and shows the sample size used for each method and measure. The amount of missing data is the difference between the number of participants included in each method and measure and the original sample from which it was drawn.

**Educational attainment.** We used administrative data from the Norwegian National Educational Database, classified according to the Norwegian Standard Classification of Education (NUS2000), to identify individuals' educational activities and backgrounds. This standard is

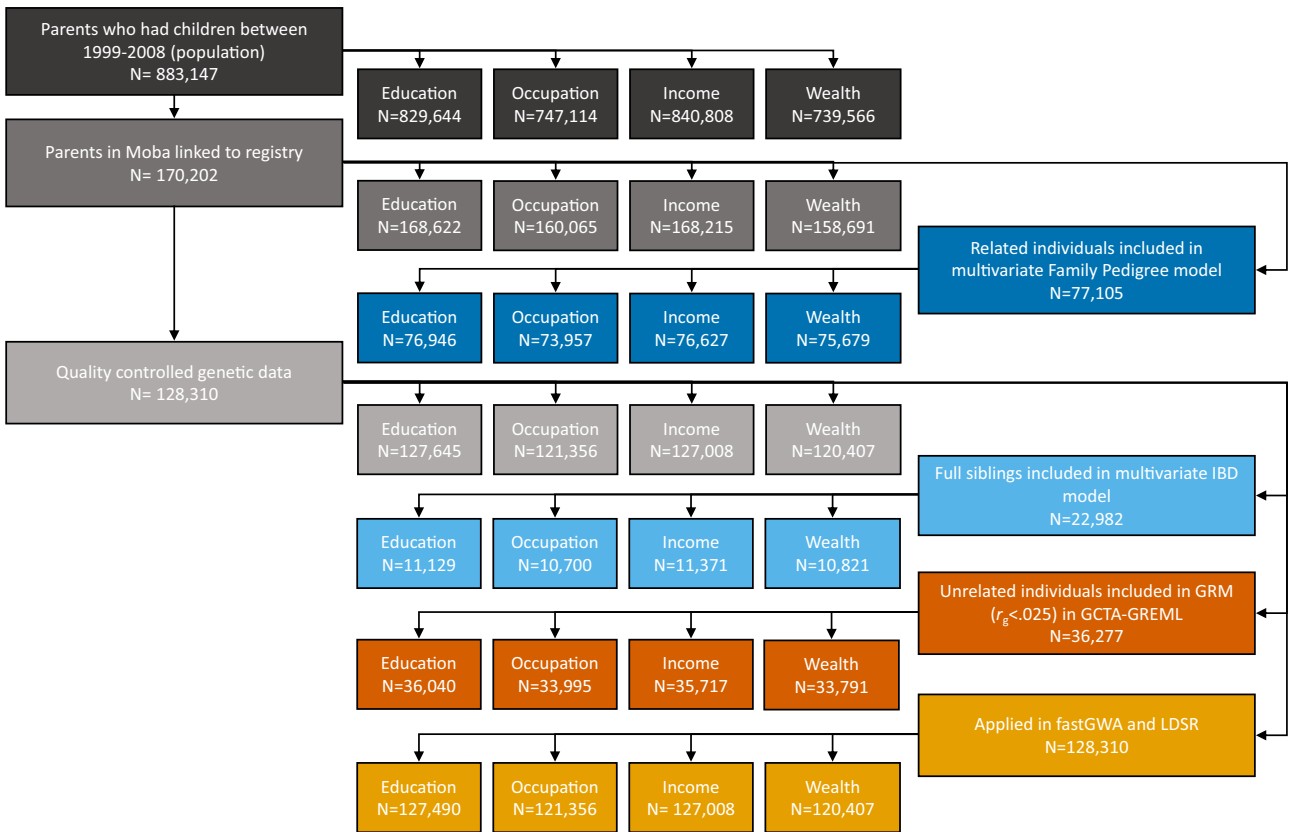

**Fig. 6 | Sample flowchart.** Flowchart of the population, MoBa parents, quality-controlled genetic data, the number of linked registry observations, and the sample size of each applied method.

used in Statistic Norway's education statistics and other statistics where education is included as a variable. More information can be found at: http://www.ssb.no/en/utdanning/norwegian-standard-classification-of-education. The educational attainment data was formatted in the International Standard Classification of Education 2011 and converted to the number of years required in Norway to achieve each level. The number of years was used as it provides a more intuitive way to interpret the results and is standard in statistical genetic analyses of educational attainment. To capture the education level at a specific life stage for all participants (parenthood), we used data on the highest educational attainment recorded between ages 35 and 45.

**Occupational prestige.** Statistics Norway collected and coded the data on occupation. More information on data production can be found here: https://www.ssb.no/en/arbeid-og-lonn/sysselsetting. We converted the occupation codes to the international ISCO 88 format and from ISCO 88 format to Treiman's international prestige scale (SIOPS)[66]. The SIOPS scale was developed through international surveys asking people to assign prestige scores to various occupations. Summarizing the results, each occupation was assigned a numeric value based on their perceived prestige. The scale ranges from 1 to 100, with higher values indicating higher perceived prestige, and the current sample includes scores from 13 to 78. This scale has been shown to be relatively stable across contexts and time points and has been widely used in social mobility research[6,67,68]. In addition, this occupational prestige scale correlates highly with other occupational measures both phenotypically and genetically and relates very similarly to other SES dimensions as other indicators of occupational status[6,17,23].

**Income and wealth.** The income and wealth data we received from Statistics Norway is based on annual data from tax returns, The Tax Register, and the A-Ordning (established 2015; a digital reporting system for employers to report income and employment-related information about their employees to various government agencies). More information can be found at: http://www.ssb.no/en/inntekt-og-forbruk. An advantage of the registry data from Statistics Norway is that it is cross checked between different registries. A limitation is that income and wealth that is not reported to the various registries (i.e. tax evasion) is not included.

Due to the large distance between extreme outliers, the income and wealth data were log-transformed, after setting negative and zero values to one. Given the low number of negative values ($n = 40$ for income, $n = 0$ for wealth), this practice did not significantly affect the results. The log transformation reduces the relative distance between the observations and does so exponentially as the values get higher.

**Income.** We used the registry-based Statistics Norway measure of total income after taxes that consisted of wages, capital income, taxable and non-taxable transfers after taxes during a calendar year. To reduce measurement error, we averaged the income indicators across a 11-year period from age 35 to 45. We do not presume to capture the heritability of lifetime earnings.

**Wealth.** We used the registry-based Statistics Norway measure of gross wealth as the sum of real capital and estimated financial capital, i.e., all the financial resources a person legally has tied to their name. Again, we averaged the gross wealth across an 11-year period from age 35 to 45.

**Heritability estimation**
**Family pedigree ACE model.** The FP ACE model combines assumed genetic relatedness and measured phenotypic correlations to estimate three variance components: an additive genetic component (A), a shared environment component (C) and a residual non-shared

environmental component (E)[39]. The pedigree structure is inferred by recorded parent-child relationships in the population data and the genetic correlations ($r_g$) were set from to the expectation from the pedigree to be $r_g = 0.125$ for first cousins ($n = 49,423$), $r_g = 0.5$ for full siblings ($n = 27,399$) and dizygotic twins ($n = 147$), and $r_g = 1$ for monozygotic twins ($n = 136$). The shared environment is assumed to be a correlation of 1 between all pairs of siblings, and 0 between first cousins in the first model. In the second model, sibling correlations are set to one, while the shared environment correlation coefficient between first cousins are free parameters to be estimated in the model. Both models included sex as a covariate.

The standardized effects of A, C, and E on the phenotype P are respectively labeled a, c, and e. The variance of the additive genetics effect ($a^2$) equals the narrow sense heritability ($h^2$). The equations for variance and covariance components applied in each pair of related individuals $P_1$ and $P_2$ are:

$$Var(P) = a^2 + c^2 + e^2 \qquad (1)$$

$$Cov(P_1, P_2) = a^2 \times r_g + c^2 \times r_c \qquad (2)$$

Open Mx software and maximum likelihood estimation were used to estimate the $a^2$, $c^2$ and $e^2$ variance components.

**Identical-by-descent AE model.** The IBD design applies empirical genetic correlations in a variance component model[40]. We estimated genome-wide identical-by-descent allele sharing (i.e., empirically derived genetic correlation) in 11,491 full sibling pairs. The sample size was not large enough to power an ACE or a sib-regression design. We, therefore, opted for an AE model with an additive genetic component (A) and a residual component (E). KING software was applied to estimate the proportion of shared alleles in each sibling pair (mean number of SNPs = 232,818, SD = 905). The models included sex as a covariate. Similarly to the family pedigree design, we applied Open Mx and maximum likelihood to estimate the A and E variance in the following equations where the genetic correlation ($r_g$) equals the estimated proportion of genome IBD

$$Var(P) = a^2 + e^2 \qquad (3)$$

$$Cov(P_1, P_2) = a^2 \times r_g \qquad (4)$$

Ultimately allowing us to calculate the heritability, as variance of the additive genetics effect ($a^2$) equals the narrow sense heritability ($h^2$).

**GCTA-GREML.** Another way to apply empirical relatedness is by comparing the genetic correlation between pairs of unrelated individuals and their phenotypic correlation. We applied genome-wide complex trait analysis (GCTA) GREML to estimate narrow sense heritability[41]. The standard GCTA-GREML regression model was applied,

$$y = X\beta + Wu + \varepsilon \qquad (5)$$

Where $\beta$ is the effect of the covariates ($X$), $u$ is the effect of the standardized genotype matrix $W$, and $\varepsilon$ is the residuals. The variance of $y$ is expressed as

$$Var(y) = WW'u^2 + I\sigma_{\varepsilon^2} \qquad (6)$$

The genetic relationship matrix (GRM), which is a matrix of genomic relatedness between all pairs of participants, was expressed as

$$A = WW'/N \qquad (7)$$

And $g$ was defined as a normally distributed vector of the effects of individuals with $g \sim N(0, A\sigma_g^2)$. Enabling us to express the variance of y as

$$Var(y) = A\sigma_{g^2} + I\sigma_{\varepsilon^2} \qquad (8)$$

In turn allowing the calculation of heritability,

$$h^2 = \sigma_{g^2} / (\sigma_{g^2} + \sigma_{\varepsilon^2}) \qquad (9)$$

The GCTA software was run with restricted maximum likelihood estimation (REML), a relatedness cutoff of 0.025, a minor allele frequency (MAF) cutoff of 0.01, and covariates consisting of 20 PCs, batch, age, and sex. The number of individuals after relatedness cutoff was 36,051, and the number of SNPs after MAF cutoff was 1,235,694.

**LD score regression.** In addition to GREML, we applied a non-relatedness-based SNP-heritability method to estimate the total effect of common SNPs. The LD score regression approach is based on the assumption that not all causal signals are carried equally by SNPs across the genome: variants with high LD scores are more likely to carry a signal of a causal variant. In LD score regression, a reference panel is used to measure the LD of SNPs, and the extent that LD is correlated with chi square statistics from a genetic association analysis approximates the heritability of the trait in question.

To estimate LD score-based heritability, we first performed GWAS of the SES indicators. In order to include relatives in the analysis, we used the fastGWA tool in the GCTA software, which is an ultra-efficient tool for mixed linear model-based GWAS analysis[69]. We applied the same GRM as in the GREML analysis with a similar number of individuals and matching covariates. We included a sparse GRM matrix with relatives with an $r_g > 0.05$ as a covariate. The number of SNPs used was 1,092,270.

We used the LD score regression software to run the LD score regression using summary statistics from the fastGWA[27]. The intercept represents the bias, while the slope estimates the proportion of phenotypic variance explained by both the SNPs of interest and the SNPs used to estimate the LD scores.

**Assumptions of the four heritability methods.** Here we present an overview of the assumptions made in the different designs with reference to a more in-depth article[70].

For the FP design, it is assumed that the shared environment makes an equal contribution to the phenotype of interest across sibling and cousin pairs. If this equal environment assumption is violated, heritability estimates are likely to be inflated. We partially test this assumption by letting the cousin shared environment correlation be estimated rather than assumed. FP, as all four methods do, assumes random mating. Assortative mating can inflate heritability estimates and shared environmental effects through the reduction of phenotypic variance in the offspring through mating on similar traits[71].

The IBD design relies on the assumption that the proportion of identical-by-descent segments between siblings is not affected by de novo mutations. The method also relies on the assumption that the proportion genetic correlation corresponds to genetic influences affecting the trait in question and that the additive genetic covariance corresponds to the proportion identical by descent. Breach of these assumptions may inflate or deflate the estimate, depending on the breach. Heritability estimates are likely to be inflated if shared environments are not modeled. IBD also assumes no assortative mating, which could inflate estimates.

Neither GREML nor LD score regression isolates direct genetic effects, meaning that estimates may include parental genetic effects that take effect through the environment. Indirect genetic effects are considerable for education, income, and occupational prestige[22,23,48]. GREML and LD score regression are also subject to a random mating

assumption that might bias the estimates in either direction depending on whether the mating patterns introduce more genetic similarity, decrease phenotypic variation, or introduce environmental confounders. The GREML design assumes no gene-environment correlations, as it includes only unrelated individuals. However, confounding shared environmental effects such as population stratification would inflate heritability estimates. GREML also assumes only to capture additive genetic effects of common SNPs; exclusion of rare variants may underestimate heritability. GREML assumes that the genetic correlation derived from the GRM is true and that it corresponds to the SNP effect on the phenotype, leaving the method sensitive to cases where the SNP effects are not standardized and normally distributed or there is a strong local LD structure[70].

LD score regression tackles the problem of LD in a different way, assuming that rare SNPs have larger effect sizes than common SNPs. This assumption will not hold in cases where LD scores are correlated with minor allele frequency. Another assumption is the LD reference panel. If the panel population is very different from the current population, the LD scores may not be accurate. However, genetic drift is uncorrelated with LD.

### Principal component analysis

PCA is a statistical technique used to simplify complex data sets by identifying patterns and relationships between variables[42]. It aims to reduce the dimensionality of the data while retaining most of the variation present in the original dataset. PCA does this by transforming the original variables into a new set of variables, called PCs, which are linear combinations of the original variables.

Parallel analysis is a statistical technique used to determine the number of factors or components to retain in a PCA[43]. It involves comparing the eigenvalues derived from the actual data with eigenvalues derived from simulated data generated with the same number of variables and observations as the original dataset. The premise of parallel analysis is that one should retain factors or components for which the observed eigenvalues exceed the eigenvalues obtained from the random data. This method helps to avoid overestimating the number of factors or components to retain, as it provides a more stringent and objective criterion for decision-making.

We also applied the PCAtest software (https://github.com/ethanbass/PCAtest/). PCAtest is an R package designed to perform permutation-based statistical tests[44]. These tests assess the overall significance of a PCA, determine the significance of individual PC axes, and evaluate the contributions of each observed variable to the significant axes. 10,000 permutations and 10,000 bootstrap replicates were used to build 95% confidence intervals of the PCs. For the genetic and environmental PCAs we bootstrapped samples based on the covariance matrix and created 95% confidence intervals from 10,000 bootstrapped samples.

### Reporting summary

Further information on research design is available in the Nature Portfolio Reporting Summary linked to this article.

## Data availability

Data can be accessed by anyone. Instructions for access to MoBa data from the Norwegian Institute of Public Health can be found here: https://www.fhi.no/en/studies/MoBa/for-forskere-artikler/research-and-data-access/. Instructions for access to Statistics Norway data can be found here: https://www.ssb.no/en/data-til-forskning. GWAS summary statistics are available at https://doi.org/10.5281/zenodo.15130285.

## Code availability

The code used in this study is available at https://doi.org/10.5281/zenodo.15050363.

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

## Acknowledgements

This work is a part of the European Research Council (ERC) consolidator grant GeoGen "The PsychoGeography of Intergenerational Mobility: Early Life Socioeconomic Position, Mental Health, and Educational Performance" (JCE, EME, and EY; Grant agreement No. #101045526). In Addition, the European Union supported AVH and THL ("OPENFLUX" #818420) and EY ("ESSGN" #101073237, "EQOP" #818425). This work was also supported by the Research Council of Norway (#288083, #336078, and #331640). RC was supported by the Jacobs Foundation grant no. 2023-1510-00. The Norwegian registry and MoBa data used was from the project SUBPU. The Department of Psychology, University of Oslo, is responsible for the data handling of SUBPU, a Data Protection Impact Assessment (DPIA) has been signed by the head of department, and the project manager is Eivind Ystrom. SUBPU is approved by the Committees for Medical and Health Research Ethics (#2017/2205). SUBPU has agreements with the MoBa and Statistics Norway for data linkage and usage. The data access and management costs of SUBPU are financed by the Research Council of Norway (RCN) (#336078, #288083, and #314601), the European Research Council (#101045526, #818425, #101088481, and #818420), and supported by the Department of Psychology (UiO). All data management and analyses were on the secure data "Tjeneste for Sensitive Data" (TSD) facilities, owned by the University of Oslo. Resources provided by Sigma2, the National Infrastructure for High-Performance Computing and Data Storage in Norway (UNINETT), were used for analyses (#NS9867S). The authors would like to acknowledge the work of SUBPU data managers Clara Timpe and Oda van Jole.

## Author contributions

J.C.E. conceived the idea and developed the design with contributions from E.M.E., R.C., and E.Y. J.C.E. carried out analysis with support from E.M.E. and R.C. Z.A., A.V.H., and T.H.L. contributed to the interpretation of the results. J.C.E. wrote the manuscript (incl. the supplementary information) with input from all authors. E.Y. contributed to data acquisition. E.M.E., Z.A., and E.Y. supervised the project. All authors provided critical feedback, discussed the results, and helped shape the manuscript.

## Competing interests

The authors declare no competing interests.
