## [Transparent Peer Review file · Nature Communications]

The Genetic and Environmental Composition of Socioeconomic Status in Norway

Corresponding Author: Mr Joakim Ebeltoft

Version 0:

Reviewer comments:

Reviewer #1

(Remarks to the Author)

The paper "The Genetic and Environmental Composition of Socioeconomic Status: A population-wide multi-method study in >170,000 Norwegian individuals" makes a significant contribution to the field of behavioral genetics and social science genomics, offering a statistically sound analysis. However, I believe the paper requires major revisions to enhance its impact and accessibility, particularly in the introduction and background sections. I provide my suggestions below which I hope the authors would find helpful.

The introduction currently lacks a broader context, diving too quickly into empirical and statistical details without adequately framing the importance of SES (socioeconomic status) in behavioral genetics. I encourage the authors to revise the introduction to clearly outline the research question, the gaps in the existing literature, and their specific contributions to the field. Additionally, the research questions should be explicitly mentioned in the introduction to guide the reader through the text, as the current structure is somewhat difficult to follow. Moreover, the introduction should include a discussion of why the Norwegian context is studied, highlighting what makes it particularly interesting or unique beyond just data availability. This is especially important for understanding the significance of the findings related to phenotypic correlations.

The abstract also needs revision. It is too technical and does not clearly convey the authors' contributions to the field. The authors should aim for a more accessible abstract that states the research question, methodology, key findings, and the broader implications of their work.

The results section currently starts with the data section, which I recommend removing for better flow. Additionally, the paper does not sufficiently address how sample selection may impact their estimates. The authors should clarify whether they have performed sensitivity tests to assess the robustness of their findings, particularly concerning the use of different subsamples of the MoBa dataset. If different subsamples were used, the paper should explain whether the results hold when a stable sample is employed, or provide a rationale if they do not.

One major area that requires more attention is the discussion of gene-environment interplays. The paper does not sufficiently explore how these interactions affect our understanding of the heritability of social status outcomes. This is a critical oversight, given the importance of gene-environment interactions in shaping these outcomes. I suggest the authors expand on this discussion to provide a more nuanced view of the heritability estimates presented.

I also recommend the authors include a figure that visually presents the differences in SES heritabilities they are referring to in the existing literature, including confidence intervals (CIs) if possible. This would help readers better understand the puzzle the authors refer to. Speaking of figures, Figure 1 should have a more detailed legend that explains all abbreviations, and Figure 2 could benefit from a more informative legend as well. Figure 3's confidence intervals look peculiar, and I would like to see more details on how the uncertainty was calculated. Additionally, Figure 4 might need confidence intervals if they are not already included.

The text contains some stylistic issues that should be addressed. For example, there are paragraphs that consist of only one sentence, such as "A wide range of narrow-sense heritability" in the last paragraph. This disrupts the flow of the paper.

There are also multiple typos throughout the text, such as the missing word in the sentence "gross versus ?? would" at the end of the heritability of the big four section in the discussion. The term "old people" should be replaced with a more scientifically appropriate term.

The paper frequently refers readers to supplementary information but does not specify the section, figure, or table, making it difficult to follow. This should be corrected to improve readability. I also noticed that the abbreviation for LD Score Regression is inconsistently used — LDSC in the methods section and LDSR in the main text. This inconsistency might confuse readers and should be corrected.

I suggest removing the "Summary of the Findings" title and integrating it directly into the discussion section. In the

concluding remarks, the authors state that E cannot be understood without G, but it is important to nuance this statement by acknowledging that the reverse is also true.

Finally, I recommend the authors consider developing a box of additional information, similar to Mignogna et al. (2023), to help eliminate potential confusion and address the sensitivity of relating genetics to social status outcomes. The authors should also clarify whether the ethics approval specifically covered this research question and whether additional reviews were conducted due to the sensitivity of the topic.

(Remarks on code availability)

Reviewer #2

(Remarks to the Author)

Thank you for the opportunity to review the study "The Genetic and Environmental Composition of Socioeconomic Status: A population-wide multi-method study in >170,000 Norwegian individuals" submitted to Nature Communications. This is a generally well-written and methodologically sound paper, with a major strength being the use of the same sample to apply different methods in estimating the heritability of SES across various indicators. However, I have several minor and major suggestions for the authors to consider.

Abstract:

- The statement, "Multivariate covariance analysis revealed commonality in genetic and familial environmental factors, but not in individual-specific factors," is not very clear. I suggest rephrasing for clarity.
- Another unclear phrase is, "As we minimize differences in estimates due to measurement error and sample characteristics." Clarifying the intended meaning here would improve the abstract's readability.

Introduction:

- The study focuses on adult SES, but the introduction begins with a discussion on how SES relates to child well-being and adolescent substance use. I suggest using references related to adult SES or comparing how SES relates across the entire lifespan, as there is ample evidence supporting this broader perspective.

Literature and Methodology:

- I recommend including the recent GWAS of income (<https://www.biorxiv.org/content/10.1101/2024.01.09.574865v1#:~:text=Our GWAS%2Dderived polygenic index,attributed to direct genetic effects.>). If you wish to differentiate between wealth and income, please clarify this distinction in the manuscript. Additionally, how is wealth measured in this context?
- While it is commendable that you compare estimates using different methods within the same sample, it is important to acknowledge that the heritability of SES varies globally. I suggest making it clear that your study cannot address all the limitations of existing literature.
- It might be helpful to discuss occupational prestige earlier in the manuscript. How is it typically measured? It seems like a very subjective measure, so further clarification could be beneficial.

Wording and Clarity:

- Some of the wording is unclear. For example, when discussing the "big 4," the phrase "chronologically ordered, and widely impactful features of SES" is vague. Please clarify what is meant here.
- The following passage is difficult to follow and seems to oversimplify the concept: "This misinterpretation ignores the biological axiom that organisms make a continual string of adaptive actions to navigate their ever-changing surroundings, which also extends to humans and their socioeconomic environments. The effects of genetics are always mediated through the environments they unfold in, which makes the study of genetics' role in SES differences effectively a study of society." It is important to note that this study focuses on individual differences within a subsample of the Norwegian population, not on society as a whole. Additionally, the manuscript does not measure how traits like educational attainment are valued within society. I suggest rephrasing this section.

Data and Analysis:

- The measurement of SES indicators is not clearly defined, especially since they are measured yearly. I suggest reporting the correlations between these indicators across years and indicating which specific measurement (e.g., the most recent, average) was used in the analysis for every indicator. I see that you take the highest reported measure of educational attainment, for example, but the transparency of the measures would be beneficial here (e.g. how much they change over adulthood).
- It is commendable that the authors address and report the sample's representativeness. However, have the authors considered adjusting the sample to make it more representative and then repeating the analyses? It would be interesting to see how representativeness influences heritability estimates (See <https://www.nature.com/articles/s41562-023-01579-9> or <https://academic.oup.com/ije/article/53/3/dyae054/7666749>).
- While the authors indicate that they use four different methods to estimate the heritability of SES and use the same sample, this is inaccurate. The sample size for each method varies widely. Can the authors conduct sensitivity analyses using exactly the same sample for each method, even if this reduces the sample size?
- Table 2 presents genetic correlations, but one number is missing. I suggest including all relevant correlations, such as phenotypic and environmental correlations, and presenting them across different methods.
- I do not agree that the choice of field of education is an indicator of SES. Could the authors explain this choice or consider removing it?

Conclusion:

- The manuscript would benefit from a stronger conclusion. What are the remaining unanswered questions? Do any policy or research recommendations arise from this study?

(Remarks on code availability)

Version 1:

Reviewer comments:

Reviewer #1

(Remarks to the Author)

I left my comments after authors' responses:

Reviewer #1 (Remarks to the Author):

The paper "The Genetic and Environmental Composition of Socioeconomic Status: A population-wide multi-method study in >170,000 Norwegian individuals" makes a significant contribution to the field of behavioral genetics and social science genomics, offering a statistically sound analysis. However, I believe the paper requires major revisions to enhance its impact and accessibility, particularly in the introduction and background sections. I provide my suggestions below which I hope the authors would find helpful.

Thank you for highlighting the contribution of the paper and providing useful feedback on how to enhance its impact and accessibility. We appreciate the opportunity and believe that the heavily revised introduction and background sections are far improved.

The introduction currently lacks a broader context, diving too quickly into empirical and statistical details without adequately framing the importance of SES (socioeconomic status) in behavioral genetics. I encourage the authors to revise the introduction to clearly outline the research question, the gaps in the existing literature, and their specific contributions to the field. Additionally, the research questions should be explicitly mentioned in the introduction to guide the reader through the text, as the current structure is somewhat difficult to follow. Moreover, the introduction should include a discussion of why the Norwegian context is studied, highlighting what makes it particularly interesting or unique beyond just data availability. This is especially important for understanding the significance of the findings related to phenotypic correlations.

Thank you for the suggestions. We have revised the whole introduction in line with the Reviewer's comments as follows:

1) Clearly outlining the three key research questions

"The research questions in the current study are:

1. What are the genetic, shared environmental, and non-shared environmental contributions to socioeconomic status indicators in a Norwegian context?
2. How much do 'big four' heritability estimates vary between genotype and family based methods?
3. How much of the genetic, shared environmental, and non-shared environmental contributions are shared across educational attainment, occupational prestige, income, and wealth?" [Introduction, p. 5]

2) Describing current gaps in prior research:

"Heritability is a population statistic that is conditional on the features of the population and is inherently reflecting the characteristics of the populations we study." [Introduction, p. 3] ... "To assess differences in heritability across SES indicators and methods, the use of a consistent sample is needed." [Introduction, p. 4]

"The approaches are historically divided into, first, family-based methods that infer genetic associations by applying expected relatedness between twins and other family members, and, second, genotype methods that estimate genetic associations between unrelated individuals using empirical variations in single nucleotide polymorphisms (SNP) in the genome. There is often a gap between family-based and unrelated genotype-based heritability estimates referred to as 'missing heritability.' This gap has also been demonstrated in genetically informed studies of SES." ... "To assess the severity of the 'missing heritability' problem for SES, we need to compare samples that share similar population characteristics." [Introduction, p. 3]

"A crucial question in the familial origins of SES is the structure of the influences on the different outcomes." ... "In sum, there is a need for multivariate studies on the 'big four' SES indicators in a consistent sample, using a variety of methods and objective measures." [Introduction, p. 4]

3) Describing how our study contributes in light of the gaps:

"For all analyses, we draw subsamples from the same population-wide cohort of parents largely born in post-welfare Norway and use yearly objective registry data from between ages 35 and 45." [Introduction, p. 5]

"First, this allows us to compare the relative importance of genetic and environmental factors for the 'big four' in Norway, a

social-democratic universal welfare state.” [Introduction, p. 5]

“We apply four different heritability methods drawn from the same sample to enable comparison of estimates across methods, two based on the principles of quantitative genetic approaches and two based on molecular genetic approaches (Table 1).” [Introduction, p. 4]

“Second, we evaluate the commonality of genetic and environmental factors of the SES ‘big four’ by performing multivariate covariance analysis and applying a dimensionality reducing technique.” [Introduction, p. 5]

4) Describing features of the Norwegian context:

“We conduct our analysis on a Norwegian sample, and there are several elements to be aware of when studying genetic and environmental variation in Norway. First, Norway has systems with social safety nets that reduces impact of negative life events on SES, and creates opportunities for second chances in socioeconomic attainment 32. Second, Norway and other social-democratic welfare states possess features that reduce the importance of certain environmental factors 33. For example, universal high-quality pre-school systems relieve the burden of financial resources as well as time that can be spent in the labor market. Third, intergenerational mobility is relatively high in Norway, partly due to its social-democratic welfare state that provides universal access to social protection and higher education 32,34. Higher intergenerational mobility has been found to be positively correlated with heritability 35. One interpretation of this correlation is that when differences created by environmental barriers or opportunities (such as the cost of higher education) are minimized, genetic differences account for relatively more of the variation in outcomes 36. Fourth, despite the strong level of educational and income equality in Norway, wealth disparities are relatively high and have increased over time 37,38. Lastly, the availability of high-quality, yearly Norwegian register data (Table 3) enhances the accuracy of findings as it eliminates biases associated with self-reported questionnaire measures 28.” [Introduction, p. 6]

These are great revisions that have improved the text.

The abstract also needs revision. It is too technical and does not clearly convey the authors' contributions to the field. The authors should aim for a more accessible abstract that states the research question, methodology, key findings, and the broader implications of their work.

We have revised the abstract to be less technical. We tried making it more accessible, while highlighting the aims, methodology, key findings, and the broader implications. For example, we cut "Multivariate covariance analysis revealed commonality in genetic and familial environmental factors, but not in individual-specific factors," and replaced it with "Our results also highlight considerable common influences on the four SES indicators among genetic and shared environmental factors, but not in non-shared environmental factors." [Abstract, p. 2]

In the sentence no. 3 ('We use...'), I suggest replacing the word 'perform' with 'employ' because methods cannot be performed.

The results section currently starts with the data section, which I recommend removing for better flow. We have removed the data section to the beginning of the supplementary information.

Great

Additionally, the paper does not sufficiently address how sample selection may impact their estimates.

Thank you for highlighting this important point. We addressed this issue in the limitations at the end of the discussion: "First, it has limited generalizability. Educational attainment, occupational prestige, income, and wealth are upwardly biased in the current sample compared to the broader population (see Supplementary Table S1). Therefore, the estimates might be more applicable to a middle-class to upper-middle-class Norwegian population. Future research should address the bias introduced by nonrandom study participation 26." [Discussion, p. 21]

We also expanded Table S1 of the supplementary information to include a wider range of descriptives for the population, the MoBa sample, and for the four sub-samples used in the study.

I am happy with the revisions.

The authors should clarify whether they have performed sensitivity tests to assess the robustness of their findings, particularly concerning the use of different subsamples of the MoBa dataset. If different subsamples were used, the paper should explain whether the results hold when a stable sample is employed, or provide a rationale if they do not. Thank you for pointing this out. We now say that we analysed individuals from a "consistent sample" rather than the "same sample". Unfortunately, we cannot use exactly the same samples for each method. The samples are often mutually exclusive. For example, GCTA-GREML assumes that the individuals are unrelated (we used a cutoff of genetic relatedness of <2.5%), while the family pedigree and IBD designs rely on first cousins and siblings (genetic relatedness >12%). All possible sibling-pairs were used for IBD, and they were a subset of the Family pedigree sample. IBD is a design based on pairs of siblings (Visscher, 2006), excluding cousins from the design. Applying the Family Pedigree design with only the siblings from the IBD method wouldn't be possible, as the Family Pedigree design uses assumed genetic correlation ($r=.5$ for full siblings) and there would not be variation to enable comparison (such as MZ vs. DZ or siblings vs. cousins).

As for GREML and the GWA+LDSC designs, we performed a GWA with the same sample as the GREML. Using the same

sample for GREML (~36,000) as opposed to our full genotype sample of ~128,000, GWA+LDSC heritability estimates dropped from 19.9% to 6.7% for education, 15% to 5.3% for occupation, 6.8% to 2% for income, and 8% to 2.4% for wealth. GWAS requires a large sample size to gain a large amount of significant hits. Applying the same sample for GREML and GWA+LDSC might make sense for sample sizes >100,000, depending on the phenotype, but for sample sizes of around ~36,000, the properties of the two methods are too different to compare heritability estimates.

To highlight the differences between the subsamples we provided descriptives for men and women within each subsample in supplementary Table S1. We added the following sentences to the limitations: "Second, due to the inherent properties of the designs, such as the reliance upon related or unrelated individuals, the same subsample cannot be used for all four methods. Supplementary Table S1 shows slight differences between the subsamples, yet these are far smaller than those between the overall population and the main sample." [Discussion, p. 21]

The revisions further strengthened the analytical side of the paper and provide a clearer, more transparent picture for the reader.

One major area that requires more attention is the discussion of gene-environment interplays. The paper does not sufficiently explore how these interactions affect our understanding of the heritability of social status outcomes. This is a critical oversight, given the importance of gene-environment interactions in shaping these outcomes. I suggest the authors expand on this discussion to provide a more nuanced view of the heritability estimates presented.

Thank you for highlighting this. As you point out, the gene-environment interactions are key in shaping these outcomes. We added the following section in the discussion to address this: "To the extent that genetic factors contribute to SES, they do so probabilistically through gene-environment correlations and interaction mechanisms. For example, as women's opportunities for education strengthened, the gender differences in the relationship between genetics and education weakened 56. In Norway genetic influences on educational performance are moderated by schools 57. Proximal and distal social factors—ranging from parents to institutions and macroeconomic events—play crucial roles in shaping individual socioeconomic outcomes through the cumulative facilitation or suppression based on genetically influenced traits." [Discussion, p. 19]

Great

I also recommend the authors include a figure that visually presents the differences in SES heritabilities they are referring to in the existing literature, including confidence intervals (CIs) if possible. This would help readers better understand the puzzle the authors refer to.

This is a great idea. We created a figure representing heritability estimates in the existing literature. We focused the selection of papers around 1) covering all four indicators, 2) most recent and reliable studies, 3) studies that covered multiple indicators, 4) limit number of papers to adhere to recommended citation limit.

This is an important figure and a great addition. One minor suggestion is to increase the font size for better readability.

Speaking of figures, Figure 1 should have a more detailed legend that explains all abbreviations, and Figure 2 could benefit from a more informative legend as well.

Thank you for pointing this out. We have made more detailed legends in Figure 1 and Figure 2 in the Result section.

These are great.

Figure 3's confidence intervals look peculiar, and I would like to see more details on how the uncertainty was calculated. Thank you for noticing the error with matching CIs to data points in the plotting code. Please see the corrected Figure 4. For how the CIs were calculated see here for the estimation and here for the plotting.

I am glad it is now corrected.

Additionally, Figure 4 might need confidence intervals if they are not already included.

Thank you for addressing this. We applied PCAtest (<https://github.com/arleyc/PCAtest>) with 10000 random permutations and 10000 bootstrap replicates to build 95% CIs for the phenotypic PCA. For the genetic and environmental PCAs we bootstrapped samples based on the covariance matrix and created 95% CIs from 10000 bootstrapped samples. Code for this available on the GitHub of first author.

This is also great.

The text contains some stylistic issues that should be addressed. For example, there are paragraphs that consist of only one sentence, such as "A wide range of narrow-sense heritability" in the last paragraph. This disrupts the flow of the paper.

We agree about the importance of harmonising the style and formatting. We made several changes to improve style and flow. For example, we removed the headers in the Introduction. Each paragraph consists of multiple sentences highlighting a point.

Thank you for revising the text accordingly.

There are also multiple typos throughout the text, such as the missing word in the sentence "gross versus ?? would" at the end of the heritability of the big four section in the discussion.

All authors have meticulously read through the document to correct for typos.

Thank you for revising the text accordingly.

The term "old people" should be replaced with a more scientifically appropriate term.

This sentence has now been removed.

Great.

The paper frequently refers readers to supplementary information but does not specify the section, figure, or table, making it difficult to follow. This should be corrected to improve readability.

Thank you for pointing this out. All references to the supplementary information now specify the relevant figures and tables.

It is now much easier to follow.

I also noticed that the abbreviation for LD Score Regression is inconsistently used — LDSC in the methods section and LDSR in the main text. This inconsistency might confuse readers and should be corrected.

We addressed this issue by only using LD Score Regression as an abbreviation.

Great.

I suggest removing the "Summary of the Findings" title and integrating it directly into the discussion section.

We have now removed the "Summary of the Findings" title and integrated it directly into the discussion section.

Great.

In the concluding remarks, the authors state that E cannot be understood without G, but it is important to nuance this statement by acknowledging that the reverse is also true.

We have addressed this lack of nuance by adding the sentence: "To understand the relatively high heritability estimates for education and occupational prestige, we must consider the environment under investigation: Norway's robust universal welfare state with free education and a comprehensive safety net."

I agree with this amendment.

Finally, I recommend the authors consider developing a box of additional information, similar to Mignogna et al. (2023), to help eliminate potential confusion and address the sensitivity of relating genetics to social status outcomes.

This is a very good suggestion. We added a box of additional information before the result section called "Ethical considerations." This hopefully addresses the contentiousness of the topic.

"The nature-nurture debate has been and will remain contentious. The history of genetic research is tainted by the eugenic movement and linked human rights abuses. It is crucial to acknowledge this history while also recognizing that genetically informed tools can help us better understand complex relationships such as those between socioeconomic inequalities and health. Appropriate interpretation of heritability and environmental estimates requires both scientific and ethical sensitivity.

In this paper, we estimate the genetic and environmental variance components of four aspects of socioeconomic status in Norway. Heritability estimates reflect additive genetic effects but do not specify the mechanisms of these effects, which are likely to operate via environmental channels 42. Further interpretation of heritability estimates requires well-specified models that include gene-environment correlations and interactions. If genes and environments do not combine additively to shape behavior but mainly interact, the model is incorrectly specified and may result in misleading inferences regarding the importance of genes and environments.

Genetic determinism, essentialism, and reductionism are common interpretational pitfalls with ethical implications 43. First, genetic factors do not determine but probabilistically influence SES outcomes in concert with social factors. In Norway, due to free and open education and universal study stipends and loans, genetic factors may explain more of the variance in educational attainment than in contexts where environmental factors such as family education, networks, and wealth restrict educational opportunities. Not just historically but recently, genetic determinism has been misused to support atrocities such as eugenics and racially motivated violence 44. Emphasizing that genetics do not determine people's socioeconomic outcomes is therefore paramount. Second, an individual's genetic 'endowment' for SES does not constitute their underlying essence. Some research suggests that essentialist beliefs can lead to prejudice. Third, reductionism is the misinterpretation that our genetic makeup can mechanistically explain higher-level phenomena like SES. A classic example is the 'red hair hypothetical' 45. If children with red hair are not allowed in school, their literacy will be worse, and the genes that determine red hair will, in this specific societal context, be associated with illiteracy. However, the genetic signal for red hair, which has a straightforward biological pathway, becomes incorrectly assigned a severely warped social pathway, and the genetic signal is wrongly interpreted as a significant cause of illiteracy." [Introduction, p. 7]

The authors raised important points here and provided a transparent message, which I am pleased with.

The authors should also clarify whether the ethics approval specifically covered this research question and whether additional reviews were conducted due to the sensitivity of the topic.

Thank you for bringing attention to the ethics. Additional reviews have been conducted: "The project has undergone review by independent ethics advisors appointed by the European Research Council (Grant agreement No. 101045526)." [Ethics, p. 31]

Thank you for clarifying and it is also an important amendment to the text.

(Remarks on code availability)

Reviewer #2

(Remarks to the Author)

Thank you for thoroughly addressing the reviewer's comments; this has greatly improved the manuscript's readability. Congratulations on this excellent paper!

(Remarks on code availability)

Dear Reviewers,

Thank you for considering the manuscript for revision. Your comments and suggestions have substantially improved the article. We give our point-by-point responses and changes below. Alterations are either shown with tracked changes or highlights where whole paragraphs have been rewritten or introduced.

Reviewer #1 (Remarks to the Author):

The paper “The Genetic and Environmental Composition of Socioeconomic Status: A population-wide multi-method study in >170,000 Norwegian individuals” makes a significant contribution to the field of behavioral genetics and social science genomics, offering a statistically sound analysis. However, I believe the paper requires major revisions to enhance its impact and accessibility, particularly in the introduction and background sections. I provide my suggestions below which I hope the authors would find helpful.

Thank you for highlighting the contribution of the paper and providing useful feedback on how to enhance its impact and accessibility. We appreciate the opportunity and believe that the heavily revised introduction and background sections are far improved.

The introduction currently lacks a broader context, diving too quickly into empirical and statistical details without adequately framing the importance of SES (socioeconomic status) in behavioral genetics. I encourage the authors to revise the introduction to clearly outline the research question, the gaps in the existing literature, and their specific contributions to the field. Additionally, the research questions should be explicitly mentioned in the introduction to guide the reader through the text, as the current structure is somewhat difficult to follow. Moreover, the introduction should include a discussion of why the Norwegian context is studied, highlighting what makes it particularly interesting or unique beyond just data availability. This is especially important for understanding the significance of the findings related to phenotypic correlations.

Thank you for the suggestions. We have revised the whole introduction in line with the Reviewer’s comments as follows:

1) Clearly outlining the three key research questions

“The research questions in the current study are:

1. What are the genetic, shared environmental, and non-shared environmental contributions to socioeconomic status indicators in a Norwegian context?
2. How much do ‘big four’ heritability estimates vary between genotype and family based methods?
3. How much of the genetic, shared environmental, and non-shared environmental contributions are shared across educational attainment, occupational prestige, income, and wealth?” [Introduction, p. 5]

2) Describing current gaps in prior research:

“Heritability is a population statistic that is conditional on the features of the population and is inherently reflecting the characteristics of the populations we study.” [Introduction, p. 3] ...

“To assess differences in heritability across SES indicators and methods, the use of a consistent sample is needed.” [Introduction, p. 4]

“The approaches are historically divided into, first, family-based methods that infer genetic associations by applying expected relatedness between twins and other family members, and, second, genotype methods that estimate genetic associations between unrelated individuals using empirical variations in single nucleotide polymorphisms (SNP) in the genome. There is often a gap between family-based and unrelated genotype-based heritability estimates referred to as ‘missing heritability.’ This gap has also been demonstrated in genetically informed studies of SES.” ... “To assess the severity of the ‘missing heritability’ problem for SES, we need to compare samples that share similar population characteristics.” [Introduction, p. 3]

“A crucial question in the familial origins of SES is the structure of the influences on the different outcomes.” ... “In sum, there is a need for multivariate studies on the ‘big four’ SES indicators in a consistent sample, using a variety of methods and objective measures.” [Introduction, p. 4]

3) Describing how our study contributes in light of the gaps:

“For all analyses, we draw subsamples from the same population-wide cohort of parents largely born in post-welfare Norway and use yearly objective registry data from between ages 35 and 45.” [Introduction, p. 5]

“First, this allows us to compare the relative importance of genetic and environmental factors for the ‘big four’ in Norway, a social-democratic universal welfare state.” [Introduction, p. 5]

“We apply four different heritability methods drawn from the same sample to enable comparison of estimates across methods, two based on the principles of quantitative genetic approaches and two based on molecular genetic approaches (Table 1).” [Introduction, p. 4]

“Second, we evaluate the commonality of genetic and environmental factors of the SES ‘big four’ by performing multivariate covariance analysis and applying a dimensionality reducing technique.” [Introduction, p. 5]

4) Describing features of the Norwegian context:

“We conduct our analysis on a Norwegian sample, and there are several elements to be aware of when studying genetic and environmental variation in Norway. First, Norway has systems with social safety nets that reduces impact of negative life events on SES, and creates opportunities for second chances in socioeconomic attainment³². Second, Norway and other social-democratic welfare states possess features that reduce the importance of certain environmental factors³³. For example, universal high-quality pre-school systems relieve the burden of financial resources as well as time that can be spent in the labor market. Third, intergenerational mobility is relatively high in Norway, partly due to its social-democratic welfare state that provides universal access to social protection and higher education^{32,34}. Higher intergenerational mobility has been found to be positively correlated

with heritability ³⁵. One interpretation of this correlation is that when differences created by environmental barriers or opportunities (such as the cost of higher education) are minimized, genetic differences account for relatively more of the variation in outcomes ³⁶. Fourth, despite the strong level of educational and income equality in Norway, wealth disparities are relatively high and have increased over time ^{37,38}. Lastly, the availability of high-quality, yearly Norwegian register data (Table 3) enhances the accuracy of findings as it eliminates biases associated with self-reported questionnaire measures ²⁸.” [Introduction, p. 6]

The abstract also needs revision. It is too technical and does not clearly convey the authors' contributions to the field. The authors should aim for a more accessible abstract that states the research question, methodology, key findings, and the broader implications of their work. We have revised the abstract to be less technical. We tried making it more accessible, while highlighting the aims, methodology, key findings, and the broader implications. For example, we cut "Multivariate covariance analysis revealed commonality in genetic and familial environmental factors, but not in individual-specific factors," and replaced it with "Our results also highlight considerable common influences on the four SES indicators among genetic and shared environmental factors, but not in non-shared environmental factors." [Abstract, p. 2]

The results section currently starts with the data section, which I recommend removing for better flow.

We have removed the data section to the beginning of the supplementary information.

Additionally, the paper does not sufficiently address how sample selection may impact their estimates.

Thank you for highlighting this important point. We addressed this issue in the limitations at the end of the discussion: "First, it has limited generalizability. Educational attainment, occupational prestige, income, and wealth are upwardly biased in the current sample compared to the broader population (see Supplementary Table S1). Therefore, the estimates might be more applicable to a middle-class to upper-middle-class Norwegian population. Future research should address the bias introduced by nonrandom study participation ²⁶." [Discussion, p. 21]

We also expanded Table S1 of the supplementary information to include a wider range of descriptives for the population, the MoBa sample, and for the four sub-samples used in the study.

The authors should clarify whether they have performed sensitivity tests to assess the robustness of their findings, particularly concerning the use of different subsamples of the MoBa dataset. If different subsamples were used, the paper should explain whether the results hold when a stable sample is employed, or provide a rationale if they do not.

Thank you for pointing this out. We now say that we analysed individuals from a "consistent sample" rather than the "same sample". Unfortunately, we cannot use exactly the same samples for each method. The samples are often mutually exclusive. For example, GCTA-GREML assumes that the individuals are unrelated (we used a cutoff of genetic relatedness of <2.5%), while the family pedigree and IBD designs rely on first cousins and siblings (genetic relatedness >12%). All possible sibling-pairs were used for IBD, and they were a subset of the Family pedigree sample. IBD is a design based on pairs of siblings (Visscher,

2006), excluding cousins from the design. Applying the Family Pedigree design with only the siblings from the IBD method wouldn't be possible, as the Family Pedigree design uses assumed genetic correlation ($r=.5$ for full siblings) and there would not be variation to enable comparison (such as MZ vs. DZ or siblings vs. cousins).

As for GREML and the GWA+LDSC designs, we performed a GWA with the same sample as the GREML. Using the same sample for GREML (~36,000) as opposed to our full genotype sample of ~128,000, GWA+LDSC heritability estimates dropped from 19.9% to 6.7% for education, 15% to 5.3% for occupation, 6.8% to 2% for income, and 8% to 2.4% for wealth. GWAS requires a large sample size to gain a large amount of significant hits. Applying the same sample for GREML and GWA+LDSC might make sense for sample sizes >100,000, depending on the phenotype, but for sample sizes of around ~36,000, the properties of the two methods are too different to compare heritability estimates.

To highlight the differences between the subsamples we provided descriptives for men and women within each subsample in supplementary Table S1. We added the following sentences to the limitations: "Second, due to the inherent properties of the designs, such as the reliance upon related or unrelated individuals, the same subsample cannot be used for all four methods. Supplementary Table S1 shows slight differences between the subsamples, yet these are far smaller than those between the overall population and the main sample." [Discussion, p. 21]

One major area that requires more attention is the discussion of gene-environment interplays. The paper does not sufficiently explore how these interactions affect our understanding of the heritability of social status outcomes. This is a critical oversight, given the importance of gene-environment interactions in shaping these outcomes. I suggest the authors expand on this discussion to provide a more nuanced view of the heritability estimates presented.

Thank you for highlighting this. As you point out, the gene-environment interactions are key in shaping these outcomes. We added the following section in the discussion to address this: "To the extent that genetic factors contribute to SES, they do so probabilistically through gene-environment correlations and interaction mechanisms. For example, as women's opportunities for education strengthened, the gender differences in the relationship between genetics and education weakened⁵⁶. In Norway genetic influences on educational performance are moderated by schools⁵⁷. Proximal and distal social factors—ranging from parents to institutions and macroeconomic events—play crucial roles in shaping individual socioeconomic outcomes through the cumulative facilitation or suppression based on genetically influenced traits." [Discussion, p. 19]

I also recommend the authors include a figure that visually presents the differences in SES heritabilities they are referring to in the existing literature, including confidence intervals (CIs) if possible. This would help readers better understand the puzzle the authors refer to.

This is a great idea. We created a figure representing heritability estimates in the existing literature. We focused the selection of papers around 1) covering all four indicators, 2) most recent and reliable studies, 3) studies that covered multiple indicators, 4) limit number of papers to adhere to recommended citation limit.

Speaking of figures, Figure 1 should have a more detailed legend that explains all abbreviations, and Figure 2 could benefit from a more informative legend as well. Thank you for pointing this out. We have made more detailed legends in Figure 1 and Figure 2 in the Result section.

Modelled variance components

- A
- ▲ AE
- ACE

Method

- Family Pedigree
- Identity-by-descent
- GCTA-GREML
- LD score regression

First cousin shared environment correlations applied:

- ▲ Assumed shared environment correlation between first cousins of 0
- Estimated shared environment correlation between first cousins of .59 (.04)

Figure 3's confidence intervals look peculiar, and I would like to see more details on how the uncertainty was calculated.

Thank you for noticing the error with matching CIs to data points in the plotting code. Please see the corrected Figure 4. For how the CIs were calculated see here for the estimation and here for the plotting.

Additionally, Figure 4 might need confidence intervals if they are not already included. Thank you for addressing this. We applied PCAtest (<https://github.com/arleyc/PCAtest>) with 10000 random permutations and 10000 bootstrap replicates to build 95% CIs for the phenotypic PCA. For the genetic and environmental PCAs we bootstrapped samples based on the covariance matrix and created 95% CIs from 10000 bootstrapped samples. Code for this available on the GitHub of first author.

The text contains some stylistic issues that should be addressed. For example, there are paragraphs that consist of only one sentence, such as "A wide range of narrow-sense heritability" in the last paragraph. This disrupts the flow of the paper.

We agree about the importance of harmonising the style and formatting. We made several changes to improve style and flow. For example, we removed the headers in the Introduction. Each paragraph consists of multiple sentences highlighting a point.

There are also multiple typos throughout the text, such as the missing word in the sentence "gross versus ?? would" at the end of the heritability of the big four section in the discussion. All authors have meticulously read through the document to correct for typos.

The term "old people" should be replaced with a more scientifically appropriate term. This sentence has now been removed.

The paper frequently refers readers to supplementary information but does not specify the section, figure, or table, making it difficult to follow. This should be corrected to improve readability.

Thank you for pointing this out. All references to the supplementary information now specify the relevant figures and tables.

I also noticed that the abbreviation for LD Score Regression is inconsistently used — LDSC in the methods section and LDSR in the main text. This inconsistency might confuse readers and should be corrected.

We addressed this issue by only using LD Score Regression as an abbreviation.

I suggest removing the "Summary of the Findings" title and integrating it directly into the discussion section.

We have now removed the "Summary of the Findings" title and integrated it directly into the discussion section.

In the concluding remarks, the authors state that E cannot be understood without G, but it is important to nuance this statement by acknowledging that the reverse is also true.

We have addressed this lack of nuance by adding the sentence: "To understand the relatively high heritability estimates for education and occupational prestige, we must consider the environment under investigation: Norway's robust universal welfare state with free education and a comprehensive safety net."

Finally, I recommend the authors consider developing a box of additional information, similar to Mignogna et al. (2023), to help eliminate potential confusion and address the sensitivity of relating genetics to social status outcomes.

This is a very good suggestion. We added a box of additional information before the result section called "Ethical considerations." This hopefully addresses the contentiousness of the topic.

"The nature-nurture debate has been and will remain contentious. The history of genetic research is tainted by the eugenic movement and linked human rights abuses. It is crucial to acknowledge this history while also recognizing that genetically informed tools can help us better understand complex relationships such as those between socioeconomic inequalities and health. Appropriate interpretation of heritability and environmental estimates requires both scientific and ethical sensitivity.

In this paper, we estimate the genetic and environmental variance components of four aspects of socioeconomic status in Norway. Heritability estimates reflect additive genetic effects but do not specify the mechanisms of these effects, which are likely to operate via environmental channels⁴². Further interpretation of heritability estimates requires well-specified models that include gene-environment correlations and interactions. If genes and environments do not combine additively to shape behavior but mainly interact, the model is incorrectly specified and may result in misleading inferences regarding the importance of genes and environments.

Genetic determinism, essentialism, and reductionism are common interpretational pitfalls with ethical implications⁴³. First, genetic factors do not determine but probabilistically influence SES outcomes in concert with social factors. In Norway, due to free and open education and universal study stipends and loans, genetic factors may explain more of the variance in educational attainment than in contexts where environmental factors such as family education, networks, and wealth restrict educational opportunities. Not just historically but recently, genetic determinism has been misused to support atrocities such as eugenics

and racially motivated violence⁴⁴. Emphasizing that genetics do not determine people's socioeconomic outcomes is therefore paramount. Second, an individual's genetic 'endowment' for SES does not constitute their underlying essence. Some research suggests that essentialist beliefs can lead to prejudice. Third, reductionism is the misinterpretation that our genetic makeup can mechanistically explain higher-level phenomena like SES. A classic example is the 'red hair hypothetical'⁴⁵. If children with red hair are not allowed in school, their literacy will be worse, and the genes that determine red hair will, in this specific societal context, be associated with illiteracy. However, the genetic signal for red hair, which has a straightforward biological pathway, becomes incorrectly assigned a severely warped social pathway, and the genetic signal is wrongly interpreted as a significant cause of illiteracy." [Introduction, p. 7]

The authors should also clarify whether the ethics approval specifically covered this research question and whether additional reviews were conducted due to the sensitivity of the topic. Thank you for bringing attention to the ethics. Additional reviews have been conducted: "The project has undergone review by independent ethics advisors appointed by the European Research Council (Grant agreement No. 101045526)." [Ethics, p. 31]

Reviewer #2 (Remarks to the Author):

Thank you for the opportunity to review the study "The Genetic and Environmental Composition of Socioeconomic Status: A population-wide multi-method study in >170,000 Norwegian individuals" submitted to Nature Communications. This is a generally well-written and methodologically sound paper, with a major strength being the use of the same sample to apply different methods in estimating the heritability of SES across various indicators. However, I have several minor and major suggestions for the authors to consider. Thank you for the positive and constructive feedback.

Abstract:

- The statement, "Multivariate covariance analysis revealed commonality in genetic and familial environmental factors, but not in individual-specific factors," is not very clear. I suggest rephrasing for clarity.

Thank you for this specific feedback. We have revised the statement as follows: "Our results also highlight considerable common influences on the four SES indices among genetic and shared environmental factors, but not in non-shared environmental factors." [Abstract, p.2]

- Another unclear phrase is, "As we minimize differences in estimates due to measurement error and sample characteristics." Clarifying the intended meaning here would improve the abstract's readability.

Thank you for addressing the readability with a specific example. We have revised the statement as follows: "By drawing subsamples from a consistent sample and using registry-based data, we reduce differences in estimates due to population characteristics and measurement error." [Abstract, p. 2]

Introduction:

- The study focuses on adult SES, but the introduction begins with a discussion on how SES relates to child well-being and adolescent substance use. I suggest using references related

to adult SES or comparing how SES relates across the entire lifespan, as there is ample evidence supporting this broader perspective.

We have now added references that are related to adult outcomes associated with SES.

“It is well established that socioeconomic status (SES) is connected to an array of important life outcomes, such as health and mortality ¹, and subjective well-being ².”

Literature and Methodology:

- I recommend including the recent GWAS of income

([https://www.biorxiv.org/content/10.1101/2024.01.09.574865v1#:~:text=Our GWAS%20derived polygenic index,attributed to direct genetic effects.](https://www.biorxiv.org/content/10.1101/2024.01.09.574865v1#:~:text=Our%20GWAS%20derived%20polygenic%20index,attributed%20to%20direct%20genetic%20effects.)).

We previously cited Kweon et al. (2024) in the introduction (in the first paragraph under “Wide range of narrow sense heritability”). With Figure 1 we better capture some of the main findings in the current literature, including Kweon et al.’s estimates.

If you wish to differentiate between wealth and income, please clarify this distinction in the manuscript. Additionally, how is wealth measured in this context?

Thank you for pointing out that this was not clear. Register data on income and wealth share the same sources. To make the distinction clearer we have described what they have in common under subheading “Financial registry data from Statistics Norway” and differentiated the income and wealth measures in two separate subheadings:

“Register data on income and wealth

The income and wealth data we received from Statistics Norway is based on annual data from tax returns, The Tax Register, and the A-Ording (established 2015; a digital reporting system for employers to report income and employment-related information about their employees to various government agencies). More information: Income and wealth statistics for households – SSB. An advantage of the registry data from Statistics Norway is that it is cross checked between different registries. A limitation is that income and wealth that is not reported to the various registries (i.e. tax evasion) is not included.

Due to the large distance between extreme outliers, the income and wealth data was log transformed. Negative and zero values were first set to one. Given the low number of negative values (n = 40 for income, n = 0 for wealth), this does not significantly skew the results. The log transformation reduces the distance between the observations and does so exponentially as the values get higher. This reduces the relative distance between the observations.

Income

We used the registry-based Statistics Norway measure of total income after taxes that consisted of wages, capital income, taxable and non-taxable transfers after taxes during a calendar year. To reduce measurement error, we averaged the income indicators across a 11-year period from age 35 to 45. We do not presume to capture the heritability of lifetime earnings.

Wealth

We used the registry-based Statistics Norway measure of gross wealth as the sum of real capital and estimated financial capital, i.e., all the financial resources a person legally has tied to their name. Again, we averaged the gross wealth across an 11-year period from age 35 to 45." [Methods, p. 25]

- While it is commendable that you compare estimates using different methods within the same sample, it is important to acknowledge that the heritability of SES varies globally. I suggest making it clear that your study cannot address all the limitations of existing literature.

We agree that this is an important point. Although the paragraph following the subheading "Heritability is contingent on the population" addresses the first sentence of this point, it might not come across that our study does not address the heritability of other populations. We underlined this point by addressing the context specificity throughout the text, from the title ("...Composition of Socioeconomic Status **in Norway**") to the discussion.

"Heritability is a population statistic that is conditional on the features of the population and is inherently reflecting the characteristics of the populations we study." [Introduction, p. 3]

"...compare the relative importance of genetic and environmental factors for the 'big four' in Norway, a social-democratic universal welfare state." [Introduction, p. 5]

"We conduct our analysis on a Norwegian sample, and there are several elements to be aware of when studying genetic and environmental variation in Norway." [Introduction, p. 6]

"We comprehensively show the relative genetic and environmental contributions to SES in Norway ..." [Discussion, p. 17]

"First, it has limited generalizability. Educational attainment, occupational prestige, income, and wealth are upwardly biased in the current sample compared to the broader population

(see Supplementary Table S1). Therefore, the estimates might be more applicable to a middle-class to upper-middle-class Norwegian population. ” [Discussion, p. 21]

“To understand the relatively high heritability estimates for education and occupational prestige, we must consider the environment under investigation: Norway’s robust universal welfare state with free education and a comprehensive safety net.” [Discussion, p. 21]

- It might be helpful to discuss occupational prestige earlier in the manuscript. How is it typically measured? It seems like a very subjective measure, so further clarification could be beneficial.

The occupational prestige scale has been shown to be relatively stable across contexts and time points and has been widely used in social mobility research. We added a table in the introduction with a description of the measure on occupational prestige, which will provide the reader enough background to understand the results, before delving deeper into the measure in the Methods section. We also wrote a bit more in the methods section on the specific occupational prestige measure and how it relates to other occupational prestige measures.

Table 3: The ‘big four’ SES indicators and how they were measured in the current study. For further description, see Method section under Measures.

SES indicator	Current measure
Educational attainment	Data from Norwegian National Educational Database was formatted in the International Standard Classification of Education (ISCED) 2011 and converted to the number of years required in Norway to obtain each category. We used the highest educational attainment registered between age 35 and 45.
Occupational prestige	Occupational codes from Statistics Norway were converted to Treiman’s Standard International Occupational Prestige Scale (SIOPS) ³⁹ . SIOPS has been shown to be relatively stable across contexts and time points and has been widely used in social mobility research ^{6,40,41} . We averaged occupational prestige across a 11-year period from age 35 to 45.
Income	Statistics Norway’s cross-referenced registry-based measure of an individual’s total income after taxes, which consists of wages, capital income, taxable and non-taxable transfers after taxes during a calendar year. We used log transformed and averaged income across a 11-year period from age 35 to 45.
Wealth	Statistics Norway’s cross-referenced registry-based measure of an individual’s gross wealth, which is a sum of real capital and estimated financial capital, i.e., all the financial resources a person legally has tied to their name. We used log transformed and averaged wealth across a 11-year period from age 35 to 45.

"In addition, this occupational prestige scale correlates highly with other occupational measures both phenotypically and genetically, and relates very similarly to other SES dimensions as other indicators of occupational status ^{6,17,23}." [Methods; Measures; Occupational Prestige]

Wording and Clarity:

- Some of the wording is unclear. For example, when discussing the "big 4," the phrase "chronologically ordered, and widely impactful features of SES" is vague. Please clarify what is meant here.

We agree. We have removed this sentence and edited the paragraph about the 'big four' to be more succinct:

"While there is broad agreement on the significance of SES, there is less consensus on how to operationalize and measure the construct across multiple disciplines ³⁻⁶. For example, in a recent literature review, SES was represented using 149 unique indicators ⁴. Each of the indicators captures a specific aspect of socioeconomic variation, while also reflecting a common status dimension ^{5,7-10}. Researchers typically study this commonality by calculating an SES composite index based on several indicators ¹¹. Most operationalizations of SES make use of at least one or more of the 'big four' SES indicators – educational attainment, occupational prestige, income, and wealth ^{4,12}. However, a systematic test and comparison of their shared or specific environmental and genetic sources of variation on the 'big four' remains largely lacking." [Introduction, p. 2]

- The following passage is difficult to follow and seems to oversimplify the concept: "This misinterpretation ignores the biological axiom that organisms make a continual string of adaptive actions to navigate their ever-changing surroundings, which also extends to humans and their socioeconomic environments. The effects of genetics are always mediated through the environments they unfold in, which makes the study of genetics' role in SES differences effectively a study of society." It is important to note that this study focuses on individual differences within a subsample of the Norwegian population, not on society as a whole. Additionally, the manuscript does not measure how traits like educational attainment are valued within society. I suggest rephrasing this section.

We have focused the discussion and revised the whole paragraph to be more specific regarding these points:

"Although individuals with similar genetics may have similar outcomes, there is no straightforward causal relationship between genetics and SES. There are three main interpretive points to consider. First, genetic factors do not determine socioeconomic outcomes ⁴³. Heritability does not imply immutability and variance explained does not equal the range of phenotypic potential afforded by one's genetics. To the extent that genetic factors contribute to SES, they do so probabilistically through gene-environment correlations and interaction mechanisms. For example, as women's opportunities for education strengthened, the gender differences in the relationship between genetics and education weakened ⁵⁶. In Norway, genetic influences on educational performance are moderated by schools ⁵⁷. Proximal and distal social factors—ranging from parents to institutions and macroeconomic events—play crucial roles in shaping individual socioeconomic outcomes through the cumulative facilitation or suppression based on genetically influenced traits. A

second pitfall is interpreting heritability as a fixed value. Cultural, economic, and political changes are constantly occurring, making heritability estimates a snapshot that can vary over time and context. Lastly, heritability is a population statistic and cannot be applied to individuals.”

Data and Analysis:

- The measurement of SES indicators is not clearly defined, especially since they are measured yearly.

Thank you for addressing this. To address this issue we created an SES indicator table in the introduction (Table 2) with the measurement definition. We also added more information about the SES indicators in the method section.

I suggest reporting the correlations between these indicators across years and indicating which specific measurement (e.g., the most recent, average) was used in the analysis for every indicator. I see that you take the highest reported measure of educational attainment, for example, but the transparency of the measures would be beneficial here (e.g. how much they change over adulthood).

The correlation between highest obtained educational level achieved between ages 35 and 45 and the highest achieved educational level regardless of age was $r=.95$. Intraclass correlation for occupational prestige was $.80$, for income was $.13$, and for wealth $.76$. The low ICC for income indicates that the average is not very representative for each individual year. This information was added to the Supplementary Information under *Descriptive information about the Norwegian Mother, Father, and Child Cohort*.

- It is commendable that the authors address and report the sample's representativeness. However, have the authors considered adjusting the sample to make it more representative and then repeating the analyses? It would be interesting to see how representativeness influences heritability estimates (See <https://www.nature.com/articles/s41562-023-01579-9> or <https://academic.oup.com/ije/article/53/3/dyae054/7666749>).

Thank you for bringing up the representativeness of the sample. Weights as suggested in the links for the MoBa sample are not yet available. We therefore investigated the representativeness and compared the number of children, age at first birth, cohabitation, year of birth, educational attainment, occupational prestige, income, and wealth across the population, MoBa, and the four subsamples. The results are in supplementary Table S1 for transparency regarding the samples and subsamples.

- While the authors indicate that they use four different methods to estimate the heritability of SES and use the same sample, this is inaccurate. The sample size for each method varies widely. Can the authors conduct sensitivity analyses using exactly the same sample for each method, even if this reduces the sample size?

Thank you for pointing this out. We now say that we analysed individuals from a “consistent sample” rather than the “same sample”. Unfortunately, we cannot use exactly the same samples for each method. The samples are often mutually exclusive. For example, GCTA-GREML assumes that the individuals are unrelated (we used a cutoff of genetic relatedness of $<2.5\%$), while the family pedigree and IBD designs rely on first cousins and siblings (genetic relatedness $>12\%$). All possible sibling-pairs were used for IBD, and they were a

subset of the Family pedigree sample. IBD is a design based on pairs of siblings (Visscher, 2006), excluding cousins from the design. Applying the Family Pedigree design with only the siblings from the IBD method wouldn't be possible, as the Family Pedigree design uses assumed genetic correlation ($r=.5$ for full siblings) and there would not be variation to enable comparison (such as MZ vs. DZ or siblings vs. cousins).

As for GREML and the GWA+LDSC designs, we performed a GWA with the same sample as the GREML. Using the same sample as for GREML (~36,000) as opposed to ~127,000, GWA+LDSC heritability estimates dropped from 19.9% to 6.7% for education, 15% to 5.3% for occupation, 6.8% to 2% for income, and 8% to 2.4% for wealth. GWAS requires a large sample size to gain a large amount of significant hits. Applying the same sample for GREML and GWA+LDSC might make sense for sample sizes >100,000, depending on the phenotype, but for sample sizes of around ~36,000, the properties of the two methods are too different to compare heritability estimates.

To highlight the differences between the subsamples we provided descriptives for men and women within each subsample in supplementary Table S1. We added the following sentences to the limitations: "Second, due to the inherent properties of the designs, such as the reliance upon related or unrelated individuals, the same subsample cannot be used for all four methods. Supplementary Table S1 shows slight differences between the subsamples, yet these are far smaller than those between the overall population and the main sample." [Discussion, p. 21]

- Table 2 presents genetic correlations, but one number is missing. I suggest including all relevant correlations, such as phenotypic and environmental correlations, and presenting them across different methods.

Thank you for the suggestion. We expanded the table to include all phenotypic, genetic, and environmental correlations across methods.

Table 4: Phenotypic, genetic, shared environmental, and non-shared environmental correlations across all four methods. Standard errors (SE) in parenthesis.

Phenotypic correlation			
	Education	Occupation	Income
Occupation	.58 (0.002)		
Income	.24 (0.002)	.38 (0.002)	
Wealth	.27 (0.002)	.29 (0.002)	.37 (0.002)
Family pedigree, genetic correlation			
Occupation	.47 (.03)		
Income	.35 (.05)	.59 (.03)	
Wealth	.90 (.02)	.58 (.03)	.39 (.05)
Identical-by-descent, genetic correlation			
Occupation	.59 (.03)		
Income	.59 (.02)	.73 (.03)	

Wealth	.93 (.01)	.69 (.03)	.66 (.03)
GCTA-GREML, genetic correlation			
Occupation	.96 (.02)		
Income	.71 (.06)	.79 (.05)	
Wealth	.92 (.04)	.79 (.05)	.92 (.04)
LD score regression, genetic correlation			
Occupation	.95 (.04)		
Income	.81 (.04)	.90 (.05)	
Wealth	.83 (.04)	.82 (.05)	.83 (.06)
Family pedigree, shared environmental correlation			
Occupation	1 (SE<.001)		
Income	1 (SE<.001)	1 (SE<.001)	
Wealth	1 (SE<.001)	1 (SE<.001)	1 (SE<.001)
Family pedigree, non-shared environment correlation			
Occupation	.07 (.02)		
Income	.05 (.03)	.16 (.02)	
Wealth	.30 (.02)	.28 (.01)	.12 (.02)
Identity-by-descent, non-shared environmental correlation			
Occupation	-.01 (.03)		
Income	-.12 (.03)	.11 (.02)	
Wealth	.08 (.04)	.23 (.02)	-.02 (.03)

• I do not agree that the choice of field of education is an indicator of SES. Could the authors explain this choice or consider removing it?

Field of education is not widely considered an indicator of SES. However, length of education does not account for all the variability within education. For example, within the same level of educational attainment different fields have different outcomes (Kirkeboen et al., 2016; *Field of Study, Earnings, and Self-Selection*). Horizontal stratification measures such as field of education could add nuance to traditional vertical axes such as length of education. We replaced the words “socioeconomic indicator” with “horizontally differentiable indicator.” [Discussion, p. 20]

Conclusion:

• The manuscript would benefit from a stronger conclusion. What are the remaining unanswered questions? Do any policy or research recommendations arise from this study? We agree that we could emphasize the contributions and suggest more future research and policy relevance. Throughout the discussion we have added more suggestions for future research. In the first and last paragraph of the discussion, we have formulated stronger conclusions.

“Although observational and non-causal, our study provides insights into the extent of the connection between genetic variation and SES in Norway. Future research could contribute to the literature on the differences in the role of genetics across various contexts.”

[Discussion, p. 22]

“Our results are relevant to a broader audience interested in SES and the role of shared environments, demonstrating relatively larger shared environmental effects for educational attainment and wealth.” [Discussion, p. 22]

“Furthermore, the findings have implications for how we fundamentally understand and model SES, showing that the common approach of operationalizing SES as a single composite index fails to capture individual-specific influences.” [Discussion, p. 22]